# PIP$_2$ modulates TRPC3 activity via TRP helix and S4-S5 linker

Amy Clarke [1,4], Julia Skerjanz[2], Mathias A. F. Gsell [2], Patrick Wiedner[2], Hazel Erkan-Candag [2], Klaus Groschner [2], Thomas Stockner [1] ✉ & Oleksandra Tiapko [2,3,4] ✉

The transient receptor potential canonical type 3 (TRPC3) channel plays a pivotal role in regulating neuronal excitability in the brain via its constitutive activity. The channel is intricately regulated by lipids and has previously been demonstrated to be positively modulated by PIP$_2$. Using molecular dynamics simulations and patch clamp techniques, we reveal that PIP$_2$ predominantly interacts with TRPC3 at the L3 lipid binding site, located at the intersection of pre-S1 and S1 helices. We demonstrate that PIP$_2$ sensing involves a multistep mechanism that propagates from L3 to the pore domain via a salt bridge between the TRP helix and S4-S5 linker. Notably, we find that both stimulated and constitutive TRPC3 activity require PIP$_2$. These structural insights into the function of TRPC3 are invaluable for understanding the role of the TRPC subfamily in health and disease, in particular for cardiovascular diseases, in which TRPC3 channels play a major role.

Transient receptor potential (TRP) channels constitute a superfamily of Ca$^{2+}$-permeable cation channels, which feature striking functional and regulatory heterogeneity[1]. As for other transmembrane proteins, the function of TRP channels is intimately intertwined with the lipid environment, and TRP channels are critically controlled by lipid mediators[2–4]. In this regard, transient receptor potential canonical 3 (TRPC3) represents a paradigm for primarily lipid-gated ion channels. Activation of TRPC3 requires the direct interaction of the channel with the phospholipase C-generated lipid diacylglycerol (DAG)[5]. The interaction between TRPC3 and DAG has been extensively studied, with recent research identifying the precise localisation of the lipid within the channel complex (namely the L2 site), and revealing details of the functional consequence of DAG-TRPC3 interaction[6–9].

Notably, the function of TRPC3 has additionally been found to be sensitive to changes in the phospholipase C (PLC) substrate and DAG-precursor phosphatidylinositol-4,5-bisphosphate (PIP$_2$), which is considered a key regulator for a wide array of transporters and channels in the plasma membrane. PIP$_2$-regulated ion channels include various TRP isoforms, specifically members of the vanilloid (TRPV), and melastatin (TRPM) subfamilies[3,4]. PIP$_2$ amounts to approximately 2% of plasma membrane lipids and is localised in the inner leaflet[10]. PIP$_2$ is considered essential for the formation of certain lipid microdomains which serve the lateral segregation and cellular positioning of essential signalling molecules[11,12]. PIP$_2$ acts as both a signalling molecule in its own right, and as a precursor to two other signalling molecules, DAG and inositol-1,4,5-trisphosphate (IP$_3$), which are produced after the cleavage of PIP$_2$ by PLC. Interestingly, divergent functional consequences have been reported for PIP$_2$ interaction with TRPC channels, since it was shown to have both inhibitory and potentiating effects on the TRPC family[13–16].

Although TRPC3 and TRPC6 feature a sequence homology of about 75%, studies aimed at localising the regulatory PIP$_2$ binding site in these two channels reached inconsistent conclusions. A docking- and mutagenesis-based approach proposed that the PIP$_2$ binding site of TRPC6 is a site between the pre-S1 helix, voltage sensor-like domain (VSLD), and TRP helix[15], while a functional study suggested that the PIP$_2$ binding site of TRPC3 is localised at a site named the lipid 1 (L1) binding site, located between the pore domain and the VSLD[7,14].

In this work, we set out to identify the PIP$_2$ binding site of TRPC3 by using a combined in-silico and in-vitro approach. Exploiting the

[1]Department of Pharmacology, Medical University of Vienna, Vienna, Austria. [2]Division of Medical Physics and Biophysics, Medical University of Graz, Graz, Austria. [3]BioTechMed, 8010 Graz, Austria. [4]These authors contributed equally: Amy Clarke, Oleksandra Tiapko. ✉e-mail: thomas.stockner@meduniwien.ac.at; oleksandra.tiapko@medunigraz.at

availability of closed-state cryo-EM structures of TRPC3[8], we use unbiased coarse-grained simulations to localise the PIP$_2$ binding site to a pocket at the intersection of the VSLD and the pre-S1 helix, which we name L3 site. Through MD-guided mutagenesis followed by whole-cell and single channel patch clamp electrophysiology measurement we confirm that PIP$_2$ binds to the L3 site and characterise its positive modulatory effect on channel function. Importantly, by combining site directed mutagenesis, patch clamp recording with extensive MD simulations we can identify the hitherto unknown key elements which allow for transducing the signal from PIP$_2$ binding to the pore domain of TRPC3 channel. Our discovery reveals the pivotal role of PIP$_2$ in signal transmission, originating at the L3 binding site and propagating through a re-entrant loop following the TRP helix. This signal is further channelled to the pore domain through the salt bridge between the TRP helix and the S4-S5 linker, ultimately regulating TRPC3's activation.

## Results

### PIP$_2$ positively modulates TRPC3

In order to probe the interaction of PIP$_2$ with TRPC3, we used a combined electrophysiology and molecular dynamics simulations approach (Fig. 1a). These two techniques enabled us to probe both the macroscopic and microscopic details of PIP$_2$ interaction with TRPC3. First, we used whole-cell patch clamp electrophysiology to investigate the effect of removing PIP$_2$ on TRPC3 activity, using the direct and non-lipid activator GSK1702934A (GSK[17]; Fig. 1b–d). Application of GSK allows us to activate TRPC3 while bypassing the PLC pathway, which results in PIP$_2$ cleavage and the generation of DAG, thus ensuring that we preserve the PIP$_2$ levels in the membrane. In addition, we co-expressed TRPC3 with the pleckstrin homology (PH) domain, which sequesters PIP$_2$, but has no catalytic activity. We observed a significant reduction in the net current density through TRPC3 upon activation by GSK in presence of the PH domain (Fig. 1b). Importantly, PIP$_2$ scavenging had no impact on current-to-voltage relations (Fig. 1c), indicating that the pore remained unchanged but rather suggesting that the gating of the channel was altered by the removal of PIP$_2$. Calcium imaging analysis supported the data by showing that Ca$^{2+}$ influx was reduced in the presence of the PH domain (Fig. 1d).

Additionally, we performed experiments with a synthetic PIP$_2$ scavenging peptide PalPIP2. PalPIP2 consists of a palmitoyl group which partitions into the lipid bilayer and a positively charged stretch of amino acids which interacts with and thereby sequesters PIP$_2$. This approach was successfully employed to study the PIP$_2$-dependence of the voltage-gated K channel Kv7[18]. Preincubation of TRPC3-transfected HEK293 cells with DMSO (control) or with PalPIP2 (5 μM) for 10 min prior to the experiments revealed significant reduction in TRPC3 current initiated by GSK in cells incubated with PalPIP2 (Supplementary Fig. 1a). Thus, we observed consistent changes in channel function when cells were either exposed to PalPIP2 or genetically modified to express a PIP$_2$-scavenging PH domain (Fig. 1b). In aggregate, these data corroborate the role of PIP$_2$ acting as a positive modulator of TRPC3. Next, we extended the PalPIP2 experiments to examine the strength of the PIP$_2$-TRPC3 interaction. PalPIP2 is incorporated into the plasma membrane within a few seconds[18]. Thus, we stimulated TRPC3-expressing cells with GSK and applied PalPIP2 when the current reached its peak (Supplementary Fig. 1b). Acute scavenging of PIP$_2$ in the environment of an active TRPC3 complex by PalPIP2 (5 μM) administration did not impact channel function. This result indicates a rather stable incorporation of regulatory PIP$_2$ within the TRPC3 complex and a slow lipid exchange at the regulatory PIP$_2$ site.

Concurrently, we used a coarse-grained (CG) molecular dynamics approach to confirm that PIP$_2$ directly binds to TRPC3. The closed state structure of TRPC3 (PDB: 6CUD[8]) was embedded in a complex, asymmetric membrane with dimensions of 24 × 24 × 13 nm. PIP$_2$ was added to the inner leaflet at concentrations of 10%, 2%, and

0.5% (or 5%, 1%, and 0.25% of the total lipid content), to reflect the likely in vivo concentrations of this lipid[10]. For each concentration, five independent repeats were run with the PIP$_2$ lipids randomly placed in different starting positions around TRPC3. The simulations were run for 20 μs, during which PIP$_2$ could laterally diffuse through the membrane and bind and unbind to TRPC3. Our simulation data show that PIP$_2$ rapidly binds to TRPC3 (Fig. 1d–f), and the protein-lipid systems reach an equilibrium after approximately 5 μs. Moreover, PIP$_2$ rapidly binds to TRPC3, and at higher concentrations (10%, Fig. 1e) it displaces the negatively charged lipid phosphatidylserine (PS) (Supplementary Fig. 2). PIP$_2$ does not appear to displace phosphatidylcholine (PC) or phosphatidylethanolamine (PE) (Supplementary Fig. 2). It also does not appear to compete with cholesterol (Chol) for binding to TRPC3, consistent with previous observations which showed that Chol typically binds to TRPC3 at regions inaccessible to phospholipids[19].

### Computational analysis predicts that PIP$_2$ binds to the pre-S1/S1 nexus of TRPC3

Having confirmed that PIP$_2$ directly binds to TRPC3, we used the CG simulations to localise PIP$_2$ binding to specific sites. 2D density maps show that at a concentration of 10% of the inner leaflet lipids, PIP$_2$ binds non-specifically at multiple sites on the protein. In contrast, the density map of PC at the same concentration (Fig. 2a) shows that PC largely remains amongst the bulk lipids. At lower concentrations of PIP$_2$ (Fig. 2c, d), there are fewer non-specific binding events and PIP$_2$ density can be clearly observed at a site on TRPC3 corresponding to a groove on the protein adjacent to the S1 helix of the VSLD and the pre-S1 helix (Fig. 2e). The binding pocket is formed by a cluster of positively charged residues (R374, K377, R380, and K385) which we name the L3 site. Of importance, adjacent to the L3 site is a disordered re-entrant loop (Fig. 2e) which is not present in the 6CUD cryo-EM structure but has been resolved in subsequent structures (PDB: 5ZBG[20]). In order to ensure that the re-entrant loop does not interfere with PIP$_2$ binding, we ran additional simulations with TRPC3, and the re-entrant loop resolved in the[20] cryo-EM structure. Interestingly, in the presence of the re-entrant loop, PIP$_2$ density at the pre-S1/VSLD nexus is enhanced (Fig. 2g). Moreover, in the presence of the re-entrant loop, the four positively charged residues found on the pre-S1 and S1 helices interact almost exclusively with PIP$_2$ (Fig. 2f), suggesting that the re-entrant loop does not interfere with PIP$_2$ binding but rather enhances it. Correspondingly, all subsequent analysis is performed on the simulations where the re-entrant loop is present.

The 2D density plots do not discriminate between two possibilities: firstly, that PIP$_2$ molecules bind at the L3 site and remain bound for several microseconds; secondly, that PIP$_2$ molecules are in equilibrium with bulk lipids at this site, forming only short-term contacts before being displaced by another PIP$_2$ lipid. In order to discriminate between these two possibilities, we applied the maximum occupancy and contact duration metrics[21] (Fig. 2h, i). Maximum occupancy is the maximum length of time a single PIP$_2$ molecule is engaged with a specific residue of TRPC3. Higher values of maximum occupancy indicate longer term interactions. In contrast, contact duration is the total length of time a residue of TRPC3 is in contact with any PIP$_2$ molecule. Previously, applying these two metrics to analyse the interaction between TRPC3 and cholesterol resulted in the identification of two markedly different sites: a frequently occupied but shallow cholesterol binding site at the pre-S1 helix, and an embedded cholesterol binding site at the domain interface, where the lipid shows slow diffusion dynamics[19]. In contrast, applying this analysis to investigate the interaction between TRPC3 and PIP$_2$ revealed that both metrics flag the S1/pre-S1 nexus (L3 site) as a site that frequently interacted with PIP$_2$ (contact duration; Fig. 2i), and formed long-term contacts with PIP$_2$ molecules (maximum occupancy, Fig. 2h). This is suggestive of possible non-annular, regulatory lipid binding sites.

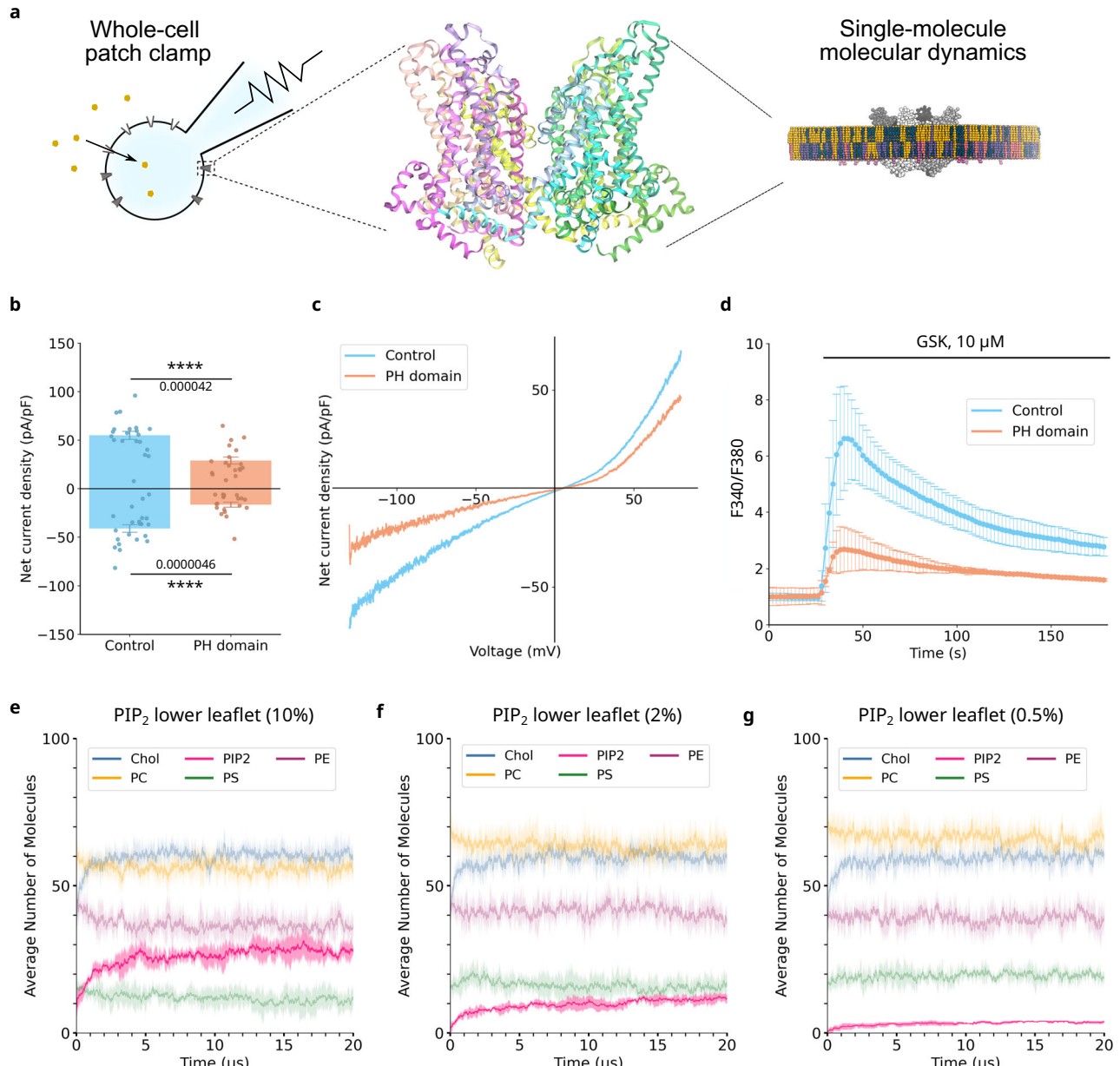

**Fig. 1 | PIP₂ positively modulates TRPC3. a** Schematic showing the tools (electrophysiology and MD simulation) used to investigate the micro- and macroscopic properties of TRPC3 interaction with PIP₂. The closed state cryo-EM structure of TRPC3 is shown in the centre (PDB: 6CUD[8]). **b** Whole-cell net current density recorded from HEK cells co-expressing TRPC3 with either empty mCherry vector (blue) or mCherry-PH domain (orange), in response to repeated applications of GSK (10 μM) at membrane potentials of −90 mV and +70 mV. Data is presented as Mean ± SEM (n = 18–20 cells). Statistical significance was assessed using a two-tailed *t* test with Bonferroni correction. **c** Representative current-voltage relationship recorded from HEK cells co-expressing TRPC3 either empty mCherry vector (blue) or mCherry-PH domain (orange), in response to repeated applications of GSK (10 μM) at membrane potentials from −130 mV to +80 mV. **d** Traces of cytosolic Ca²⁺-sensitive Fura-2 ratio recorded from HEK cells co-expressing TRPC3 either empty mCherry vector (control; blue) or mCherry-PH domain (PH domain; orange), in response to repeated applications of GSK (10 μM). GSK is applied as indicated by the bar. Data is presented as Mean ± SEM (n = 20-25 cells). Average number of molecules within 0.6 nm of TRPC3 as a function of time, for membranes containing (**e**) 10%, (**f**) 2%, and (**g**) 0.5% PIP₂ in the lower leaflet of the membrane. Five lipid species are present in the membranes: cholesterol (Chol, blue); phosphatidylcholine (PC, yellow); phosphatidylserine (PS, green); phosphatidylethanolamine (PE, purple); PIP₂ (pink). Data is averaged over 5 independent repeats and the standard deviation is shown as a lighter trace.

The size of the L3 site and the 2D density plots suggest that more than one PIP₂ molecule could be accommodated at this pocket (Fig. 2c, d, g). Indeed, in some simulations two PIP₂ molecules could be clearly observed interacting at this nexus (Supplementary Fig. 3a). In order to identify if the L3 site decomposes into two discrete binding sites, or rather represents a continuous interface where PIP₂ molecules could bind at any point along the surface, we used the phi coefficient to calculate the similarity between PIP₂ binding events, in a protocol adapted from[21]. In short, any PIP₂ molecule which bound to TRPC3 for more than 10 μs was analysed. The residues of TRPC3 which interact with the PIP₂ molecule during the binding event were identified, and then the similarity of interacting residues between different binding events was calculated using the phi coefficient (explained in full in Supplementary Fig. 3b). Agglomerative hierarchical clustering was then used to cluster PIP₂ binding events (Supplementary Fig. 3a). This analysis shows that the majority of long-term binding events occur at the L3 site, and that, of those which bind here, the majority bind at a site close to the re-entrant

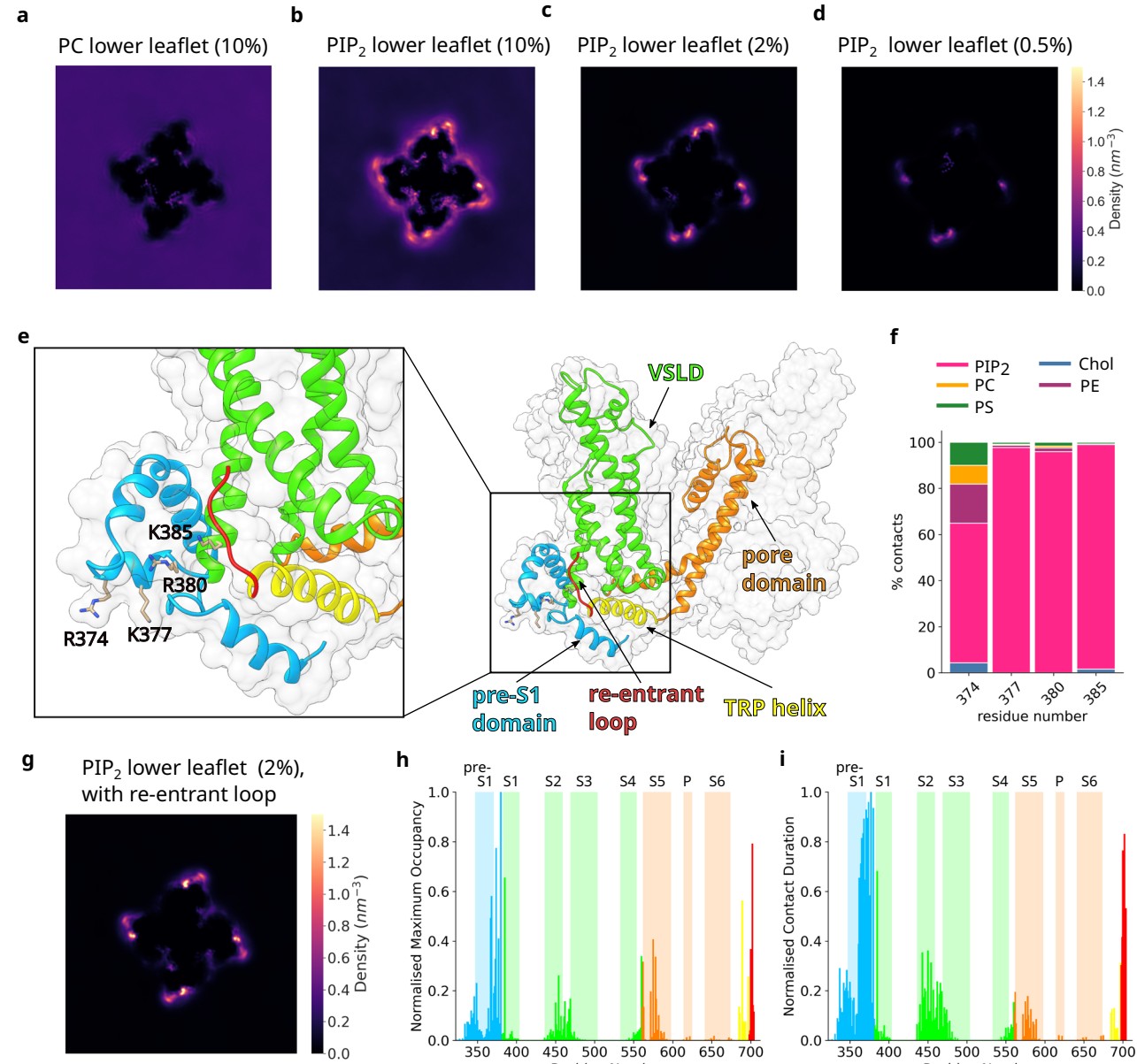

**Fig. 2 | Molecular dynamics simulations localise PIP₂ binding to a site on the VSLD. a** 2D density map showing PC density in the lower leaflet of the CG membrane. Data is averaged over 5 repeats. 2D density maps showing PIP₂ density in the lower leaflet of the CG membrane, at concentrations of (**b**) 10%, (**c**) 2%, and (**d**) 0.5%. Data is averaged over 5 repeats. **e** Structure of the transmembrane region of TRPC3 (PDB: 6CUD[8]), highlighting the key structural features: pore domain (orange), VSLD (green), pre-S1 domain (blue), TRP helix (yellow), and the re-entrant loop resolved in the[20] cryo-EM structure (red). The four positively-charged residues found at the S1/pre-S1 nexus (L3 site) are shown in stick representation. **f** The relative contribution of each lipid species to contacts with the L3 residues. Data is averaged over 5 repeats, with four monomers per repeat. The data was calculated from the equilibrated portion of the simulations (5–20 μs). **g** 2D density maps showing PIP₂ density in the lower leaflet of the membrane, at a concentration of 2%, in the presence of the re-entrant loop. **h** Normalised maximum occupancy diagrams for TRPC3 interaction with PIP₂. The data is averaged over 20 monomers. The vertical blocks indicate the transmembrane helices of TRPC3, coloured according to (**e**). The data was calculated from the equilibrated portion of the simulations (5–20 μs). **i** Normalised contact duration for TRPC3 interaction with PIP₂. The data is averaged over 20 monomers. The vertical blocks indicate the transmembrane helices of TRPC3, coloured according to (**e**). The data was calculated from the equilibrated portion of the simulations (5–20 μs).

loop. A smaller portion binds at a second, discrete site slightly farther from the re-entrant loop.

In order to gain insights how binding of PIP₂ to the periphery of TRPC3 could generate a signal which eventually could propagate towards the ion conducting pore in the centre of the channel, we took the final frame of the coarse-grained simulations and extracted the PIP₂-TRPC3 bound complex. We then embedded this complex in a smaller membrane of 15.5 × 15.5 × 13 nm, ran a short coarse-grained equilibration, and backmapped the membrane-PIP₂-TRPC3 complexes

to an all atom representation. All atom simulations of TRPC3 bound to PIP₂ were then run for 400 ns using the charmm 36 m forcefield[22,23]. As a reference, all atom simulations of TRPC3 without PIP₂ were also run. Hydrogen bonding analysis of the all atom simulations confirmed the interaction between PIP₂ and positively-charged residues of the L3 site (Supplementary Fig. 4a). Although the timescale of the simulations is too short to observe long-range conformational effects of PIP₂ binding on TRPC3 behaviour, the all atom simulations were able to capture local effects of PIP₂ binding to TRPC3. More specifically, in the

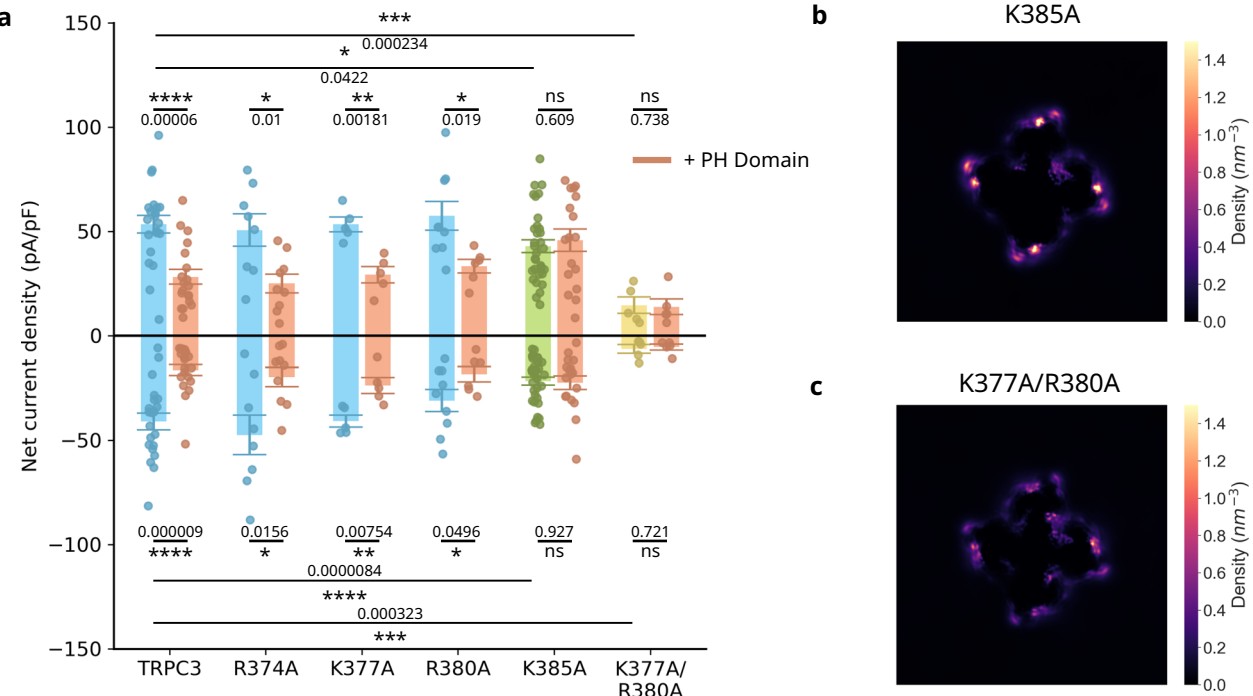

**Fig. 3 | Loss of PIP₂ binding requires the neutralisation of more than one positive charge in the L3 binding site of the TRPC3 channel. a** Whole-cell net current density recorded from HEK cells co-expressing TRPC3/L3 site mutants with either empty mCherry vector or mCherry-PH domain (orange), in response to repeated applications of GSK (10 μM) at membrane potentials of −90 mV and +70 mV. Data is presented as Mean ± SEM (*n* = 5–21 cells). Statistical significance was assessed using ANOVA followed by a two-tailed *t* test with Bonferroni correction. Not significant - ns. 2D density maps showing PIP₂ density in the lower leaflet of the CG membrane, at a concentration of 2%, for the (**b**) K385A and (**c**) K377A/R380A mutants. Data is averaged over 5 repeats.

**Electrophysiology shows that K385 has a key role in the signal transduction of PIP₂ binding**

To experimentally confirm the in-silico predictions of the PIP₂ binding site, we generated four mutants with the aim to reduce PIP₂ binding by removing the positively charged amino acids that directly interacted with PIP₂: R374A, K377A, R380A, and K385A (Fig. 3). All four mutants showed expression levels comparable to the wild-type (WT) TRPC3 (Supplementary Fig. 4a), as determined by fluorescent microscopy, confirming that the mutations did not change surface expression. Next, we used whole-cell patch clamp electrophysiology to characterise the effect of the four mutants on TRPC3 activity. The TRPC3 mutants R374A, K377A, R380A did not show functional differences compared to WT TRPC3, while K385A exhibited both an impaired basal activity (Supplementary Fig. 5b) and a reduction in activity after stimulation with GSK, suggesting that this residue is of importance for channel activity. Sequence alignment of TRPC channels revealed that K385 is fully conserved among the human TRPC channels (Supplementary Fig. 5c). To confirm the general relevance across the TRPC family of this specific residue for the TRPC family, we mutated the respective residue (K442) in TRPC6 to alanine, as TRPC6 is the closest homologous channel to TRPC3. The K442A mutant was challenged with GSK (10 μM), as GSK has the same channel activating property on TRPC3 and TRPC6, using patch clamp experiments (Supplementary Fig. 5d). Similar to K385A in TRPC3, K442A in TRPC6 showed significantly reduced response to stimulation compared to WT, supporting the notion of a conserved role among TRPC channels. Furthermore, ref. 24. also noted a substantial reduction in DAG-stimulated current when the amino acid K442 in TRPC6 was replaced

by glutamine (K442Q), underscoring the pivotal role of this particular residue in mediating activation by both lipid and synthetic agonists.

Next we tested the involvement of the mutants in PIP₂ binding by coexpression with the PIP₂ scavenging PH domain. Channel activation by GSK stimulation (Fig. 3a) showed WT-like behaviour for the R374A, K377A, and R380A mutants in presence of the PH domain (Fig. 3a), suggesting that PIP₂ remains bound to the mutant channels and exerts a comparable positive modulatory effect. In contrast, the K385A mutant exhibited no change in GSK-induced activity in presence of the PH domain, implying an important role of K385 not only in TRPC3 activity, but also in its PIP₂ sensitivity. Given that previous studies have shown that abrogation of PIP₂ binding to membrane proteins typically requires the neutralisation of more than one positive charge[25,26], we created the double mutant K377A/R380A. Fluorescent microscopy confirmed that the mutant showed expression levels comparable to WT (Supplementary Fig. 5a). Importantly, the K377A/R380A mutation showed significantly reduced net current density after stimulation with GSK (Fig. 3a) and a complete insensitivity to PIP₂ scavenging. These data suggested a loss of the positive modulatory effect of PIP₂.

The reduction in current density of the mutant channels could be caused by two reasons: firstly, PIP₂ binding at the L3 could be reduced because of the mutation or secondly, the signal transduction machinery which communicates PIP₂ binding to the pore domain could be compromised. In order to discriminate between these two possibilities, we ran coarse-grained molecular dynamics simulations of the single K385A and the double K377A/R380A mutants in the presence of 2% PIP₂ in the inner leaflet (Fig. 3b, c). 2D density maps (Fig. 3c) of K377A/R380A showed significantly reduced PIP₂ binding at the L3 site compared to the WT protein (Fig. 2c), confirming that the decline in net current density was due to a loss of PIP₂ binding. In contrast, the K385A mutant could still accumulate PIP₂ at the L3 site to levels comparable to that of the WT channel (Fig. 3b). This suggests that the reduced net current density of the K385A mutant is due to a

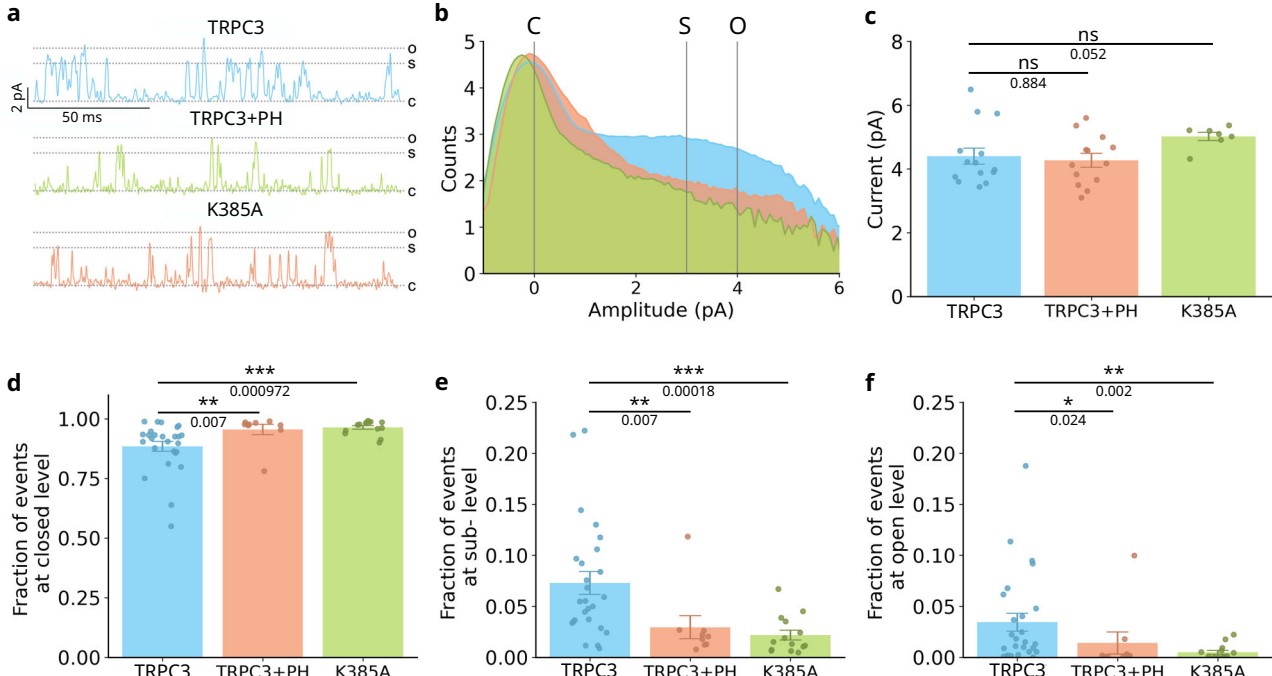

**Fig. 4 | PIP$_2$ is essential for the gating but not for the conductance of TRPC3.**
**a** Representative single-channel currents show 0.15 s of GSK (10 μM)-induced
TRPC3 (blue), TRPC3 co-expressed with PH domain (TRPC3 + PH; orange) and
K385A mutant (green) currents recorded at +80 mV in cell-attached configuration.
Closed (c), sublevels (s) and open (o) channel states are indicated. **b** Representative
logarithmic histograms of single-channel amplitude recorded from HEK cells
expressing TRPC3 (blue), TRPC3 co-expressed with the PH domain (TRPC3 + PH;
orange) or K385A mutant (green) in response to repeated applications of GSK
(10 μM) at a membrane potential of +80 mV. The histograms were fitted to sums of
Gaussian functions to determine the closed (c), sublevel (s) and open (o) ampli-
tudes. **c** Unitary currents corresponding to open (o) level recorded from HEK cells
expressing TRPC3 (blue), TRPC3 co-expressed with the PH domain (TRPC3 + PH;

orange) or K385A mutant (green) in response to repeated applications of GSK
(10 μM) at a membrane potential of +80 mV. Data is presented as Mean ± SEM
(n = 7–14 cells). Statistical significance was assessed using ANOVA followed by a
two-tailed multiple *t* test with Bonferroni correction. Not significant - ns. Fraction of
events derived from the Gausian fitting of the logarithmic histogram correspond-
ing to the closed (**d**), sub (**e**) and open (**f**) levels of single channel behaviour
recorded from HEK cells expressing TRPC3 (blue), TRPC3 co-expressed with the PH
domain (TRPC3 + PH; orange) or K385A mutant (green) in response to repeated
applications of GSK (10 μM) at a membrane potential of +80 mV. Data is presented
as Mean ± SEM (n = 7–14 cells). Statistical significance was assessed using ANOVA
followed by a two-tailed multiple *t* test with Bonferroni correction.

loss in communication between the PIP$_2$ binding site and the pore
domain. We speculate that other residues in the PIP$_2$ binding site
(R374, K377, and R374) serve to enhance binding of the PIP$_2$ molecule
by providing a sufficiently large number of positive charges in close
vicinity. The location of the four amino acids supports this proposal:
R374, K377, and R374 are found on the pre-S1 helix, an elbow-shaped
helix which projects into the lipid bilayer, whereas K385 is found at the
base of the S1 helix, in a crucial position where it can interact with
residues of the TRP helix and re-entrant loop (Fig. 2e).

Given the impact of the K385A mutant on the activity of TRPC3
and the unchanged expression level, we next explored if the reduction
in current density could be a consequence of the changes in the geo-
metry of the pore domain. Lichtenegger et al. [27] monitored the per-
meabilities of organic monovalent cations of various sizes in order to
calculate the pore diameter[27]. We employed this approach to evaluate
the pore dimensions of the K385A mutant in comparison to WT using
the whole-cell voltage clamp technique. Transfected cells were per-
fused with the physiological standard solution and stimulated with
GSK at 20 s of the recording. At the maximum current response, the
extracellular Na$^+$ (size 1.9 Å) was rapidly exchanged for different
ammonium ions (TriMA$^+$ (size ≈3 Å) and TetraMA$^+$(size ≈4 Å)). Both WT
and the K385A mutant channels (Supplementary Fig. 6a) allowed for
comparable permeation of TriMA$^+$ and TetraMA$^+$, confirming that the
K385A mutant did not lead to significant changes in the conformation
of the pore. These data are corroborated by the similarity of the in
current-voltage characteristics measured by whole cell current (Fig. 1
c). In addition, evaluation of whether the K385A mutant had an effect

on cation selectivity revealed an unaltered permeability ratio com-
pared to WT TRPC3 (TRPC3: P$_{Ca}$/P$_{Cs}$ = 1.13 ± 0.24 and K385A: P$_{Ca}$/
P$_{Cs}$ = 0.98 ± 0.16; Supplementary Fig. 6b).

To further characterise the role of the residue K385 on TRPC3
channel function, we performed single-channel recordings in HEK293
cells expressing either WT or mutant channels in the cell-attached
patch configuration. It was previously demonstrated that activation of
TRPC3 channels by a synthetic agonist is mostly associated with an
increase in the opening frequency[28]. In our measurements we detected
multiple sublevels with conductance values below the conductance of
the fully open state, suggesting a duration of the open state in the
TRPC3 channel that is shorter than the time resolution of our experi-
ments (Fig. 4a). Thus to analyse the single-channel behaviour of
TRPC3, we implemented a fraction separation of the observed events.
These events were categorised into three distinct states: the closed
state (c), the fully open state (o), and events occurring below the fully
open state, which we refer to as sublevels (s) (Fig. 4a, b). We found that
WT TRPC3 co-expressed with the PH domain (TRPC3 + PH) and the
K385A mutant exhibited no significant change in unitary conductance
compared to TRPC3 (y$_{TRPC3}$ = 55.02 ± 3.2 pS; y$_{K385A}$ = 62.73.02 ± 1.6 pS
y$_{TRPC3+PH}$ = 53.02 ± 2.7 pS; Fig. 4c). Next, we assessed the number of
events observed for the closed, sublevels, and open populations.
Notably, the K385A mutant and TRPC3 + PH fractions showed a sig-
nificant increase in the number of the events at closed state (Fig. 4d),
while the count of sublevels and opening events decreased (Fig. 4e, f).
The measured pattern of observed events suggests that modifications
to the PIP$_2$ binding site reduces the number of transitions from the

closed to the open state. From these data we conclude that $PIP_2$ binding does not affect the geometry of the pore, but instead reduces the probability of a transition from the closed to the open state.

High PalPIP2 concentrations (5–10 μM; Supplementary Fig. 7a) and a long incubation time (10 min; Supplementary Fig. 1a) were required to observe inhibitory effects on TRPC3 (Fig. 3a). Perfusion in whole-cell configuration was fairly inefficient to suppress TRPC3 activity acutely (Supplementary Fig. 1b) indicating a strong bond between TRPC3 and the lipid. Consequently, we conducted inside-out patch clamp experiments, which allow for more rapid and efficient delivery as well as removal of $PIP_2$ at the inner membrane leaflet surrounding the channel. Importantly, we observed no change in the unitary conductance of the TRPC3 channel ($y_{TRPC3} = 56.32 \pm 0.88$ pS; Supplementary Fig. 7b) compared to the cell-attached mode ($y_{TRPC3} = 55.02 \pm 3.2$ pS; Fig. 4d). However, the application of PalPIP2 (10 μM) in the inside-out patch resulted in a rapid decline in the open probability of the channel in the presence of the activator (GSK, 10 μM; Supplementary Fig. 8a). Control experiments conducted in the presence of solvent only (DMSO; 0.01%) instead of PalPIP2 demonstrated a lack of significant reductions in channel activity (Supplementary Fig. 7b, c, e).

Next, we assessed the activity and PalPIP2 sensitivity of mutants in inside-out mode. We compared K385A and two other mutations (K377A and R380A), which exhibited the highest affinity to $PIP_2$ in the MD simulations (Fig. 2f). None of the mutations significantly changed the unitary conductance of the channel ($y_{K377A} = 48,87 \pm 2,14$ pS; $y_{R380A} = 48,13 \pm 2,02$ pS; $y_{K385A} = 53,65 \pm 3,25$ pS; Supplementary Fig. 7b). The decline in the activity of K377A and R380A in the presence of PalPIP2 resembled that of TRPC3 wild type (10 μM; Supplementary Fig. 8a, c, d). Open probability of K385A was stable during perfusion with the $PIP_2$ scavenger (10 μM; Supplementary Fig. 8a, c, d) and displayed a similar lack of run down as observed in control experiments (DMSO, 0.01%; Supplementary Fig. 8b, c, e), further emphasising the significant role of this residue in $PIP_2$ sensitivity by TRPC3.

## The S4-S5 linker is a crucial element connecting the $PIP_2$ sensing domain to the gating machinery

The single channel data suggest that residue K385 does not affect the geometry of the ion conducting pore, but instead indicate that it is involved in the gating mechanism of TRPC3. Consequently, we set out to identify the pathway of signal transduction that leads to the regulation of the ion conducting pore through binding of $PIP_2$ to the L3 site. Cryo-EM structures of TRPC3 show that K385 interacts with backbone atoms of residues of the TRP helix and re-entrant loop, namely D698 and D699 and K701[8,9,20]. Our all-atom simulations confirm these interactions (Supplementary Fig. 9a, b), and found that in the presence of $PIP_2$ the number of contacts between K385 and the re-entrant loop is slightly decreased (Supplementary Fig. 9c). To test if these residues are involved in the propagation of the signal from the $PIP_2$ binding site to the re-entrant loop and the TRP helix, we created three mutants and evaluated their impact on TRPC3 activity: D698I/D699K, D698K/D699K and K701A (Supplementary Fig. 9g). None of the mutations showed a significant effect on channel activity (Supplementary Fig. 9g). Moreover, a coarse-grained computational mutant D698K/D699K accumulated $PIP_2$ at the L3 site at levels comparable to the WT protein (Supplementary Fig. 9d–f). These data show that the side chains of D698, D699, and K701 are not involved in signal propagation.

To experimentally scrutinise our functional concept in view of the difficulty to modify the protein backbone by mutagenesis, we looked for other residues which could be involved in the communication of $PIP_2$ binding from the pre-S1/S1 nexus though the TRP helix to the pore domain. Given that the all-atom simulations show a decrease in contacts between K385 and the re-entrant loop, which directly follows the TRP helix (Supplementary Fig. 9c), we explored the possibility that the TRP helix is involved in $PIP_2$-mediated channel gating. Zhao et al. proposed that full opening of the TRP ankyrin 1 in the TRPA1 channel requires allosteric coupling between the upper and lower gates[29]. Namely, the S5 helix moves upwards to widen the upper gate whilst the TRP helix initiates conformational changes in the lower gate. In addition, a coupling between the TRP helix and the linker was previously shown to be crucial for the gating in other TRP channels (TRPV4[30]; TRPM8[31]; TRPV6[32]). In four out of six cryo-EM structures (7DXB, 7DXC, 7DXE[9] and 5ZBG[20]), we observed a salt bridge between residue E684 of TRP helix and R572 residue located at S4-S5 linker (Fig. 5a, b). This salt bridge was also present in the all-atom MD simulations (Fig. 5b).

In order to probe the role of this salt bridge in TRPC3 gating, we created two single mutants, R572E and E684R, to disrupt the salt bridge that links the TRP helix and the S4-S5 linker. Neither mutant affected membrane targeting (Supplementary Fig. 10a). However, the TRP helix mutant E684R lacked basal channel activity and it could not be activated by the agonist GSK (Supplementary Fig. 10b and Fig. 5c). R572E exhibited significantly reduced basal activity (Supplementary Fig. 10b) and a lower response to GSK stimulation (Fig. 5d). Most importantly, co-expression with the PH domain was unable to reduce the activity of the R572E mutant in contrast to WT TRPC3 (Fig. 5d). To rule out that the effect was caused by the charge inversion of the R572E mutant, we created the neutralising glutamine mutant R572Q. This charge neutralisation still led to a significantly reduced GSK-induced activity of TRPC3 and, importantly, eliminated sensitivity to coexpression of the $PIP_2$ scavenging PH domain (Supplementary Fig. 10c). Results with both of the R572 mutants support the essential role of the salt bridge between R572 and E684 for the transduction of the signal originating from $PIP_2$ binding to the ion conducting pore.

To further verify the importance of the salt bridge between R572 and E684, we created the charge reversed double mutant R572E/E684R (Supplementary Fig. 10a) to explore if the TRPC3 gating function could be recovered. The double mutant was not only a regain of function compared to the functionally dead E684R mutant, but also restored sensitivity to the $PIP_2$ scavenging PH domain (Fig. 5e, f). A computational all-atom mutant R572E/E684R also retained a modest fraction of residues forming a salt bridge (Supplementary Fig. 8d) during the course of a 150 ns simulation. The regain of activity and, more importantly, recovered $PIP_2$ dependence in the R572E/E684R mutant indicates the salt bridge between TRP helix and S4-S5 linker is essential for the allosteric coupling of the lipid binding at pre-S1/S1 nexus and ion conducting pore. Taken together, our data suggest a mechanism for $PIP_2$ modulation of TRPC3 activity that involves the pre-S1 substructure, the TRP helix, and the S4-S5 linker. $PIP_2$ binds to a cluster of positively charged residues at the lipid binding site L3 at the pre-S1/S1 nexus. Binding of $PIP_2$ to the L3 triggers a sequence of structural changes, likely initiated by K385, and hinging upon its connection to the backbone of the re-entrant loop and requiring the structural integrity of the salt bridge stabilising the association between TRP helix the S4-S5 linker.

## Discussion

The signalling lipid $PIP_2$ has been long identified as a master regulator of plasma membrane channels and transporters and has been recognised to govern the function of most if not all TRP channels[2,3]. Here we present a comprehensive, mechanistic model of the $PIP_2$ sensing and gating machinery of the paradigmatic lipid sensor TRPC3. We show that $PIP_2$ controls TRPC3 by an as yet unrecognised coordination site at the pre-S1/S1 nexus (Fig. 2), where it facilitates channel opening. $PIP_2$ control over TRPC3 gating involves $PIP_2$ binding to a cluster of charged residues including K385 (Fig. 2e), which serves as a molecular switch for downstream propagation of conformational changes via the re-entrant loop and the TRP helix (Supplementary Fig. 9a–c) towards gating structures in the pore domain (Fig. 5a–c). Importantly, we unveil a pivotal regulatory role of the salt bridge connecting the TRP helix and the S4-S5 linker.

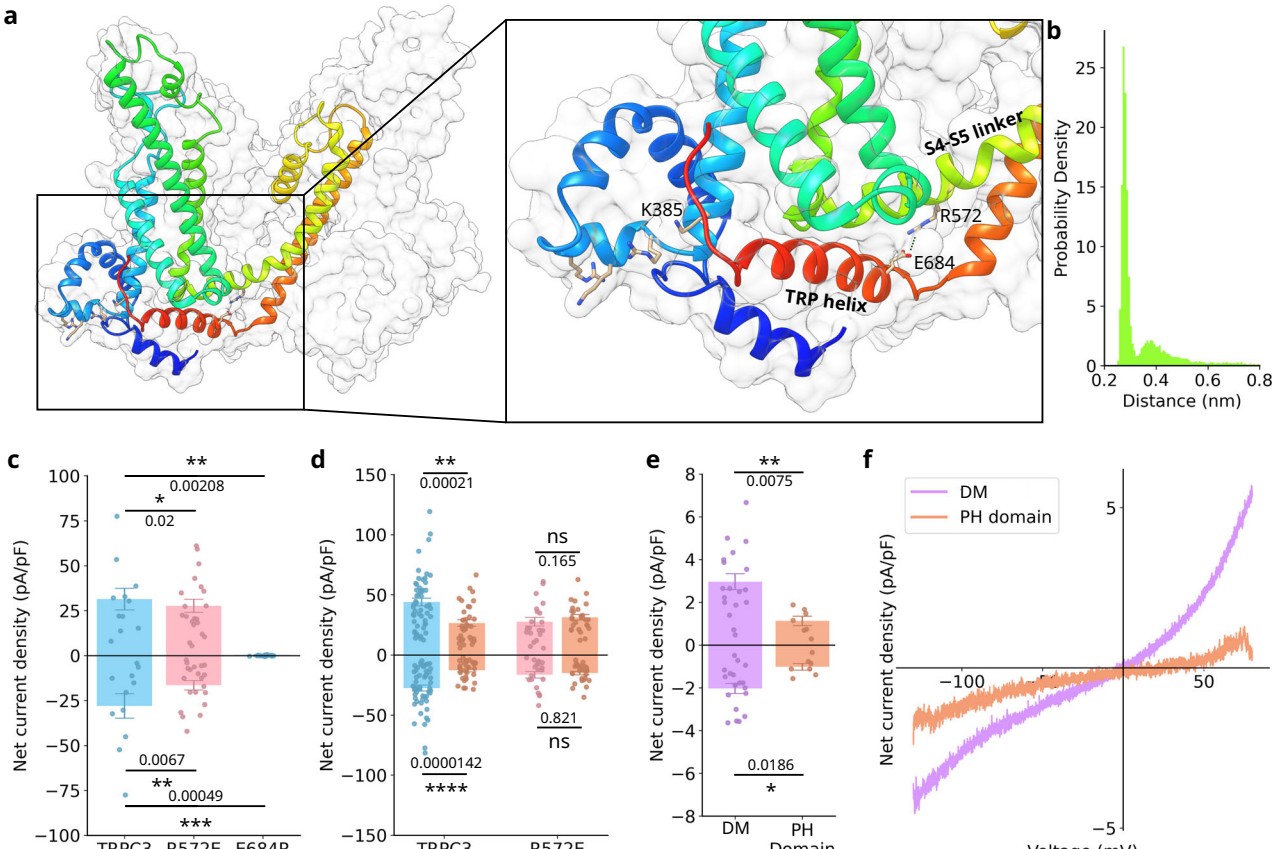

**Fig. 5 | Importance of the salt bridge between the TRP helix and S4-S5 linker for PIP$_2$ channel sensitivity. a** The structure of TRPC3 (PDB: 6CUD[8]). One monomer is shown in ribbon representation; the tetrameric shape of TRPC3 is shown in a surface representation (grey). The four positively charged residues (R374, K377, R380, K385) of TRPC3 which interact with PIP$_2$ are shown as sticks, as is the R572/E684 salt bridge. **b** Histogram showing the distance between residues R572 and E684, confirming a stable salt bridge. **c** Whole-cell net current density was recorded from HEK cells co-expressing TRPC3, R572E, and E684R mutants in response to repeated applications of GSK (10 μM) at membrane potentials of −90 mV and +70 mV. Data is presented as Mean ± SEM (n = 8–20 cells). Statistical significance was assessed using a two-tailed $t$ test with Bonferroni correction. **d** Whole-cell net current density from HEK cells co-expressing TRPC3 or R572E mutant with either empty mCherry

vector or mCherry-PH domain (orange) in response to repeated applications of GSK (10 μM) at membrane potentials of −90 mV and +70 mV. Data is presented as Mean ± SEM (n = 27–53 cells). Statistical significance was assessed using a two-tailed $t$ test with Bonferroni correction. **e** Whole-cell net current density was recorded from HEK cells co-expressing R572E/E684R double mutant (DM) with either empty mCherry vector or mCherry-PH domain (orange) in response to repeated applications of GSK (10 μM) at membrane potentials of −90 mV and +70 mV. Data is presented as Mean ± SEM (n = 7–18 cells). Statistical significance was assessed using a two-tailed $t$ test with Bonferroni correction. **f** A representative current-voltage relationship recorded from HEK cells co-expressing R572E/E684R (DM) with either empty mCherry (violet) or mCherry-PH domain (orange) in response to applications of GSK (10 μM) at membrane potentials from −130 mV to +80 mV.

Previous reports on PIP$_2$ regulation of TRPC3 provided inconsistent results and lacked mechanistic insights into the coupling between PIP$_2$ sensing and lipid-modulation of gating. By combining in-silico prediction with experimental verification, we obtained compelling evidence for a prominent PIP$_2$ binding site residing at the pre-S1/S1 nexus, a site we designate the L3 binding site. This localisation of a regulatory PIP$_2$ binding is consistent with a site recently proposed for the highly homologous TRPC6 channel[24]. Significantly, in this study, a K442Q mutation was found to result in a decrease of PIP$_2$ binding affinity about 5- to 8-fold as compared to WT TRPC6[24]. K442 is identical to the K385 residue found in TRPC3, which we propose to function both as a PIP2 binding residue and a transducer of the PIP$_2$ binding signal. Interestingly, the Mori et al. study (2022)[24], also proposes a role for residues of the distal TRP box, such as K781 and K782, in PIP$_2$ sensing. Due to lack of structural information of this region in the cryo-EM structures, we have been unable to explore whether this region plays a role in PIP$_2$ sensing by TRPC3. However, with the advent of de novo structure prediction methods such as AlphaFold, this may successfully be addressed by future research.

Another previous report suggested that PIP$_2$ binds to the L1 coordination site in TRPC3[15], which was identified as a lipid

coordination site by cryo-EM structure determination[8]. Here, Liu et al.[15] proposed that PIP$_2$ binding to the L1 binding site would inhibit TRPC3 channel opening. Our findings demonstrate a positive modulatory effect of PIP$_2$ on TRPC3 function linked to its interaction at the L3 binding site. Our results do not preclude that TRPC3 may be dually regulated by PIP$_2$, involving L3 as well as L1 site as primary recognition sites. Nonetheless, our simulation data detected only minor contact formation between PIP$_2$ and residues in the L1 region (Supplementary Fig. 3a), suggesting that affinity to the L1 would be much lower. Such a dual regulation of PIP$_2$ has been proposed for TRPV1, where depending on agonist concentration PIP$_2$ was found to act as activator or inhibitor[33]. Indeed, exploring the interplay between these two sites, in particular the relative affinity of PIP$_2$ for each site, could be an interesting future direction of research.

Our study unveiled several critical steps of the mechanism by which PIP$_2$ binding to the L3 site is translated into conformational changes of gating elements in the pore domain to govern channel activity. We present evidence supporting a multistep regulatory process involving coordinated movement of the TRP helix and the S4-S5 linker. Additionally, our findings identify K385 as a part of a lipid binding site and as a pivotal element transmitting the PIP$_2$ binding

signal to the TRP helix within the signal transduction pathway. Single-channel experiments (Fig. 4e, f) demonstrate a clear reduction in open channel probability not only in PIP$_2$ depleted conditions, but also in the presence of the K385A mutation (Fig. 4b–f). Our inside-out patch clamp experiments showed that exposure to PalPIP2 initiates a decrease in open probability in the presence of the GSK activator. Interestingly, the K385A mutation remained unaffected by the PIP$_2$ scavenger, highlighting the important role of this residue in TRPC3's sensitivity to this lipid. (Supplementary Fig. 8). Consistently, our simulation data show that the interactions of K385 to the re-entrant loop and the TRP helix are PIP$_2$-dependent (Supplementary Fig. 9a–c). This PIP$_2$-sensitive structural connection between the PIP$_2$ binding site and the TRP helix via K385 and the re-entrant loop can exert pulling forces on the TRP helix, which in turn are propagated to the ion conducting pore. This concept is supported by our experimental data, which show that the salt bridge (R572 to E684; Fig. 5a, b) between the S4-S5 linker and the TRP helix is indispensable for PIP$_2$ sensitivity of TRPC3 gating. Single point mutations which disrupt the salt bridge abrogated PIP$_2$ sensitivity (R572E: Fig. 5d and R572Q: Supplementary Fig. 10c) and the charge-reversed (R572E/E684R) double mutant at the salt bridge regained PIP$_2$ sensitivity (Fig. 5e, f).

An important question that arises from the current structural models of TRP channels and their regulatory diversity, is whether this mechanism of TRPC3 regulation could be applicable to other TRP channels. Of particular importance to advance our understanding of TRPC function is a deeper insight into the allosteric communication between the VSLD and the pore domain. Here, we propose that PIP$_2$ binding to the VSLD of TRPC3 is communicated to the pore domain via the coordinated movement of TRP helix and the S4-S5 linker. Sequence conservation within the TRPC family of the salt bridge forming residues (Supplementary Fig. 10e) suggests that the proposed signal transduction pathway is at the very least conserved within the TRPC subfamily. However, cryo-EM structures have shown that these two elements have a tight interaction in other TRP channels[31], and in TRP channels without the salt bridge, the close connection between the two is mediated by an essential hydrogen bond[30]. Similar structural rearrangements were shown for TRPM5. Ruan et al. found that the TRP helix switches its interactions from one residue of the S4-S5 linker to another in order to enable structural rearrangements in the pore domain which are necessary for opening of the channel[34]. Taken together, our observations suggest the interaction between the S4-S5 linker and TRP helix is an essential element that TRPC channels exploit as part of their regulatory arsenal. Although the mechanism may differ in specific aspects, our model developed for TRPC3 gating might therefore be generalisable for the TRP channel superfamily.

Overall, we unveiled the structural elements required for the control of TRPC3 through PIP$_2$, and provide evidence for a defined sequence of conformational events involving translating lipid sensing into a gating pattern. This interplay serves a crucial role in physiological processes within native tissues and allows for differential tuning of TRPC3 activity. The TRPC3 channel shows high constitutive activity and its basal Ca$^{2+}$ influx regulates excitability of neurons in hippocampus[35], cerebellum[36] and substantia nigra[37]. We found that PIP$_2$ regulated not only the stimulated activity, but also the constitutive activity of the TRPC3 channel (Supplementary Figs. 5b and 10b). Hence, we suggest that alterations in the physiological PIP$_2$ levels surrounding TRPC3 could act as a modulator of channel activity and, thus, regulates the intrinsic pacemaker function of this channel in neurons, highlighting the important roles of lipid-TRPC3 interactions in both health and disease.

## Methods
### Coarse-grained molecular dynamics simulations
The closed-state cryo-EM structure (PDB: 6CUD[8]) was downloaded from the Protein Data Bank (PDB)[38]. Missing residues were modelled in

**Table 1 | Lipid concentrations used in the simulations**

|        | Inner Leaflet    | Outer Leaflet |
|--------|------------------|---------------|
| PC     | 10/18/19.5/20    | 60            |
| Chol   | 30               | 30            |
| PE     | 35               | 10            |
| PS     | 15               | -             |
| PIP2   | 10/2/0.5/0       | -             |

using Modeller 9.24[39] and then the protein was truncated to residues 322–699 (isoform 3 numbering) using pdb-tools[40]. The Position of Protein in Membranes database was used to determine the orientation of TRPC3 in the membrane[41], and then the protein was converted to a coarse-grained representation using the martinize.py script (v2.4). The coarse-grained protein was then placed in a box of size 24 × 24 × 12 nm, and lipids were randomly placed in a grid-like manner around the protein using the insane.py script, according to the concentrations in Table 1 [42]. The system was solvated, neutralised, and NaCl ions were added to a final concentration of 150 mM. Five independent repeats were run.

For the simulations containing the re-entrant loop, the structure with the PDB: 5ZBG[20] was downloaded from the PDB[38]. The 6CUD and 5ZBG structures were aligned, and the re-entrant loop resolved in 5ZBG (residues 700 to 704) was added to the 6CUD structure, to create a new structure containing residues 322 to 704.

Simulations were run on the Vienna Scientific Cluster (VSC3) using GROMACS version 2020.1[43–45] with the Martini 2.2 forcefield for proteins and Martini 2.0 forcefield for lipids. The simulations were conducted at a pressure of 1 bar, maintained using the Parrinello-Rahman barostat[46], and at a temperature of 310 K, maintained using a velocity rescaling thermostat[47]. The reaction-field algorithm was used for electrostatic interactions with a cut-off of 1.1 nm, and a single cut-off of 1.2 nm was used for van der Waals interactions. The time step was 20 fs, with coordinates saved every 5000 steps. Neighbour searching was performed every 20 steps. During the production run, position restraints with a force constant of 10 were used to maintain the three-dimensional structure of TRPC3.

Analysis was carried out using scripts written in-house. VMD was used to calculate the number of lipid molecules within 0.6 nm of TRPC3. gmx densmap (GROMACS version 2020.1) was used to generate the 2D density maps. The breakdown of lipid contacts at the four positively-charged residues of the pre-S1 was calculated using gmx select, with a contact defined as two coarse-grained beads coming within 0.6 nm of each other. The maximum occupancy and contact duration metrics are derived from[21], but implemented in-house using the gmx mindist tool.

**Visualisation and image rendering were carried out using UCSF Chimera[48]**
**Clustering of PIP$_2$ binding events.** The clustering of PIP$_2$ binding events follows a methodology adapted from[21]. For each PIP$_2$ molecule in the 2% PIP$_2$ simulations, the distance between the PIP$_2$ molecule and TRPC3 was calculated as a function of time using the gmx mindist tool. If the PIP$_2$ molecule comes within 0.6 nm of TRPC3 the two are considered to be interacting. From these data, it is possible to calculate how many times each PIP$_2$ molecule interacts with TRPC3 and the length of each interaction.

To explore long-term interactions further, interactions between a PIP$_2$ molecule and TRPC3 which persisted for longer than 10 µs were then subject to further analysis (visualised in Supplementary Fig. 3a). First, gmx mindist was used to identify which specific residues of TRPC3 the PIP$_2$ molecule interacts with during the long-term binding event, using the -respertime and -printresname flags. The output of this tool is a data array with the time-resolved distances between the

PIP$_2$ molecule and each individual residue of TRPC3. From this, we created a binary data array, where a value of 1 indicates a contact between the PIP$_2$ molecule and specific residue of TRPC3 at time t. A value of 0 indicates no contact (i.e., a distance of greater than the threshold of 0.6 nm). The likelihood of two residues of TRPC3 simultaneously contacting the same PIP$_2$ molecule can then be assessed by comparing their binary arrays using the Pearson correlation coefficient implemented in the pandas.DataFrame.corr module. The output is a value ranging from -1 to +1, with -1 indicating a perfect negative correlation and +1 a perfect positive correlation. A positive value indicates the two residues show a high probability of simultaneously contacting the bound PIP$_2$ molecule, whereas a negative value indicates that binding to one residue prevents binding to the second residue. This could be the case if the two residues are not in close spatial proximity. If two residues do not contact PIP$_2$ at all during the interaction event the output of the Pearson correlation coefficient is 0.

The results were then collated into an adjacency matrix, where a positive value indicates the two residues are likely to form simultaneous contacts to the bound PIP$_2$. An adjacency matrix was created for each PIP$_2$ binding event over 10 μs. In total 29 PIP$_2$ molecules bound TRPC3 for over 10 μs. The adjacency matrices were then compared statistically using the phi coefficient implementation of scikit-learn (sklearn.metrics.matthews_corrcoef; the phi coefficient is also called the Matthews correlation coefficient). The output is a value ranging from −1 to +1. Positive values indicate the two PIP$_2$ binding events involve the same subset of TRPC3 residues whereas negative values indicate the two PIP$_2$ binding events do not involve the same subset of TRPC3 residues. This analysis was conducted pairwise for each PIP$_2$ binding event.

Finally, agglomerative hierarchical clustering was used to cluster PIP$_2$ binding events which have positive phi coefficients together. The single linkage criterion was used to create the clusters. The output is shown in Supplementary Fig. 3b. The order of the matrix was determined through agglomerative hierarchical clustering and the colour indicates the phi coefficient. In order to confirm these results, the trajectories were visually inspected to ensure that binding events clustered together correspond to the same site. Representative conformations of PIP$_2$ binding poses for group 1 and 2 are shown in Supplementary Fig. 3c.

**All atom molecular dynamics simulations.** The PIP$_2$-TRPC3 bound complexes were extracted from the final frame of the coarse-grained simulations. The complex was then embedded in a smaller membrane of 15.5 × 15.5 × 13 nm, with the same complex, asymmetric lipid composition as described above. An additional 12 residues at the N-terminus of TRPC3 were included for simulation (new protein length: residues 304–699). The smaller membranes were then equilibrated for 3 μs using the same molecular dynamics parameters described in the previous section. Subsequently, the entire complex (membrane-PIP$_2$-TRPC3) was backmapped to an all atom representation, using the backward.py script[42]. As the backmapping process can introduce small structural changes to proteins, we removed the backmapped protein and re-inserted the original 6CUD cryo-EM structure into the membrane[8]. The CHARMM36 forcefield was used to describe the system[22,23].

As the reintroduction of the cryo-EM structure can cause clashes between the lipid molecules and protein residues, the all atom simulations were subject to 100 rounds of energy minimisation, and then a short equilibration at 1 K. The short 1 K equilibration consists of two 1000-step equilibrations conducted using the md integrator, with new velocities generated each time. At this temperature, only those atoms subject to strong forces from atomic clashes will be displaced. After this, with the lipid-protein clashes resolved, the systems were equilibrated for four rounds at 310 K. Each round had consecutively

decreasing position restraints on TRPC3 (force constants of 1000, 100, 10, and 1), and was run for 2.5 ns each, giving 10 ns in total. Pressure coupling during the equilibration steps was achieved in a semi-isotropic manner using the Berendsen barostat.

The production run simulations were run for 400 ns on the Vienna Scientific Cluster (VSC) using GROMACS version 2020.1[43-45]. For the production run, pressure coupling was achieved in a semi-isotropic manner using the Parrinello-Rahman barostat[46]; temperature coupling used the velocity rescaling thermostat[47]. Protein, membrane and solvent were coupled separately. A timestep of 2 fs was used and coordinates were saved every 10 ps. The cutoff scheme was Verlet and the neighbour list was updated every 50 steps. In accordance with the CHARMM36 molecular dynamics parameters recommended for use in GROMACS, the van der Waal type was set as 'cutoff' with a force-switch vdw modifier. The switching distance (rvdw-switch) was set as 1.0 nm with the cutoff was set as 1.2 nm. The coulomb type was set as Particle Mesh Ewald (PME) with a cutoff of 1.2 nm.

Analysis was conducted using scripts written in-house, employing tools from GROMACS and MDAnalysis (Michaud-Agrawal et al.[49]; Gowers et al.[50]). The distance between residues R572 and E684 was calculated using gmx mindist. The hydrogen bonding analysis of PIP$_2$ and K385 used the Hydrogen Bond Analysis module of MDAnalysis, with the cutoff radius for a hydrogen bond set as 0.35 nm. The distance between the Cα atoms of P371 and L458 was calculated using MDAnalysis.

**Visualisation and image rendering were carried out using UCSF Chimera[48]**

**Reagents and constructs.** All reagents used were of molecular biology grade, purchased from Merck Sigma (Austria) unless specified otherwise. mCherry-PH domain (36075) and GFP-PIP5 kinase (202720) constructs purchased from AddGene constructs were provided by Roland Malli (Medical University of Graz, Graz, Austria). We cloned TRPC3 fusion construct in, pECFP-C1, and pEYFP-C1 and mCherry-C1 vectors (Clontech, Saint-Germain-en-Laye, France). Primer sequences are given in Supplementary Table 1.

**Mutagenesis.** Mutants were generated through site-directed mutagenesis using the QuickChange II Site Directed Mutagenesis Kit (Stratagene, USA). Human TRPC3 (Uniprot database ID: Q13507-3) cloned into pEYFP-C1 vector was used as template.

**Cell culture and transfection.** Human embryonic kidney 293 (HEK293) (Cell Lines Service, product number: 300192) cells were cultured in Dulbecco's Modified Eagle Medium (DMEM, D6429, Invitrogen) with 10% supplement of fetal bovine serum (FBS), HEPES (10 mmol/L), L-glutamine (2 mmol/L), streptomycin (100 μg/mL), and penicillin (100 μ/mL) at 37 °C and 5% CO$_2$ level. Cells were authenticated by short tandem repeat (STR) analysis and regular tests were performed to confirm the lack of contamination with mycoplasma. For transfection for electrophysiology and microscopy experiments, the media was aspirated and HEK293 cells were rinsed with PBS. Cells were incubated with accutase (250–500 μL) for 5 min at 37 °C. The detached cell suspension was mixed with fresh DMEM in 2:1 ratio (DMEM:accutase). 1 × 105 cells suspension was centrifuged at 300 × g for 2 min. The supernatant was discarded, and the cell pellet was suspended in serum-free medium (60 μL). Cells were transiently transfected with 1 μg plasmid DNA using PolyJet (SignaGen Laboratories) according to the manufacturer's protocol. Cells were seeded on 6 × 6 mm glass coverslips and the medium was changed after 6 h of incubation with a transfection reagent. Experiments were performed 20–24 h after transfection.

**Electrophysiology.** For whole cell analysis of TRPC3 activity, transfected HEK293 cells were seeded on glass coverslips the day before

the experiments. After 24 h, coverslips were mounted in a perfusion chamber on an inverted microscope (Zeiss Axiovert 200 M, Germany) with 40 × 0.75 objective. CoolLED pE-300 Ultra was used as an excitation source. Transfected cells were detected by illumination wavelength at 490 nm for YFP-TRPC3 and mutants and 515 nm for mCherry-PH domain. Patch-clamp recordings were performed in whole-cell configuration using an Axopatch 200B amplifier (Molecular Devices) connected with a Digidata-1550B Digitizer (Axon Instruments). The patch pipettes were pulled from thin-walled borosilicate glass (Clark Electromedical Instruments, UK) using a pipette puller (Sutter Instruments, CA, USA). Signals were low-pass filtered at 2 kHz and digitised with 8 kHz. The application of linear voltage-ramp protocols ranging from −130 to +80 mV (holding potential 0 mV) was controlled by Clampex 11.0 (Axon Instruments) software. Current densities at −90 and +70 mV were plotted against time and normalised by capacitance. For the pharmacological measurements, cells were kept at room temperature and perfused with GSK (10 μM in DMSO), PalPIP2 (5 μM in DMSO): 140 NaCl, 10 HEPES, 10 Glucose, 2 MgCl$_2$, 2 CaCl$_2$, pH adjusted to 7.4 with NaOH. Pipette solution (ICS) contained (in mM): 150 cesium methanesulfonate, 20 CsCl, 15 HEPES, 5 MgCl$_2$, 3 EGTA, titrated to pH 7.3 with CsOH. Thin-wall capillary pipettes made by borosilicate glass with filament (Harvard Apparatus, USA) were pulled to a resistance of 3–4 MΩ using a pipette puller (Sutter Instruments, CA, USA).

For PIP$_2$ scavenging with PalPIP2, cells were preincubated with either DMSO (control) or PalPIP2 (3, 5, and 10 μM) for 10 min in ECS prior to the experiments. For the rapid PIP$_2$ scavenging experiments, YFP-TRPC3 expressing cells were perfused with GSK (10 μM) starting from 30 s of the recording and when the current reached its peak, the solution was switched to GSK (10 μM)-PalPIP2 (5 μM) containing perfusion.

In order to determine permeability of organic cations, experiments were carried out in standard extracellular solution with sodium (Na$^+$, diameter d = 1.90 Å) equimolarly replaced by tri- (TriMA$^+$, d ≈ 3 Å) or tetra-methyl ammonium (TetMA$^+$, d ≈ 4 Å)[51]. In order to ensure integrity of channel function, solutions furthermore exhibited physiological concentrations of Ca$^{2+}$ and Mg$^{2+}$ (2 mM each).

For single channel analysis of TRPC3 activity, currents were recorded at RT using the Axopatch 200 B amplifier (Molecular devices, USA). Single channel activity was recorded in cell-attached or inside-out configuration. Cell-attached configuration: the bath solution contained (in mM) 145 potassium gluconate, 5.3 KCl, 3 MgCl$_2$, and 15 HEPES. Inside-out configuration: the bath solution contained (in mM) 140 potassium gluconate, 5 NaCl, 1 MgCl$_2$, 5 EGTA, and 10 HEPES. The pipette solution for both configurations contained 137 NaCl, 5 KCl, 2 CaCl$_2$, 2 MgCl$_2$, and 10 HEPES. The pH of all solutions was adjusted to 7.4. Thick-wall capillary pipettes made from borosilicate glass with filament (Harvard Apparatus, USA) were pulled using a pipette puller (Sutter Instruments, CA, USA). Patch pipettes had resistances of 20–30 MΩ for cell-attached and 10–15 MΩ for inside-out modes. The single channel activity was recorded in gap-free mode and monitored for 3–4 min. Cells were stimulated with GSK (10 μM) in presence of PalPIP2 (10 μM) or DMSO (0.01%) for cell attached mode. Single channel currents were digitised at a sampling rate of 50 kHz and filtered with the Axopatch-200B internal 4-pole low-pass Bessel filter (−3 dB cut-off at 2 kHz). The holding potential ranging from 80 mV and was controlled by the holding command function of the amplifier. Data acquisition, analysis, and further filtering with a low-pass Gaussian filter (−3 dB cut-off at 1.5 kHz) was done using pClamp11 software (Axon Instruments, Foster City, CA) and the custom made software for single channel analysis SCANA was uploaded to GitHub (https://github.com/medunigraz/scana).

To analyse the histogram of amplitude distribution in SCANA, a tool was implemented in Python V3.9.1[52] utilising the scientific computing packages NumPy V1.19.5[53] and SciPy V1.6.0[54] as well as the plotting package Matplotlib[55]. For the analysis, the following model:

$$f(x) = \sum_{i=1}^{n} \alpha_i \cdot \exp\left(-\left((x - \beta_i) \cdot \gamma_i\right)^2\right) \quad (1)$$

was implemented with parameters $\alpha_i$, $\beta_i$, and $\gamma_i$ for $i = 1, \cdots, n$ and an arbitrary but fixed $n$. To fit the model to the histogram a curve fitting method based on a non-linear least squares problem was applied to optimize the set of model parameters. From the fitted parameters $\alpha_i$, $\beta_i$, and $\gamma_i$, mean values $\mu_i = \beta_i$, standard deviations $\sigma_i = (2 \cdot \gamma_i^2)^{-1/2}$, and scaling factors $\rho_i = \sqrt{\pi} \cdot \gamma_i^{-1} \cdot \alpha_i$ can be derived and the model can be written as sum of scaled normal distributions:

$$f(x) = \sum_{i=1}^{n} \rho_i \cdot \frac{1}{\sigma_i \sqrt{2 \cdot \pi}} \exp\left(-\frac{1}{2}\left(\frac{x - \mu_i}{\sigma_i}\right)^2\right) \quad (2)$$

which simplifies the interpretation of the parameters.

**Calcium imaging.** For Ca$^{2+}$ imaging, HEK293 cells were co-transfected with YFP-TRPC3 and mCherry PH domain constructs and seeded on coverslips the day prior to the experiments. Ca$^{2+}$ changes were monitored using Fura-2 AM (Invitrogen, USA). Briefly, cells on coverslips were loaded with 1 μM Fura-2 AM for 30 min in an experimental buffer, composed of (in mM): 140 NaCl, 10 HEPES, 10 glucose, 2 CaCl$_2$, and 2 MgCl$_2$ at pH 7.4 (adjusted with NaOH). Cells which were seeded on the cover slip were transferred to a bath on an inverted microscope with 40 × 1.3 N.A. oil-immersion objective (Olympus IX71) at room temperature. During the recordings using Live Acquisition v2.6 software (FEI, Planegg, Germany), cells were excited alternately using 340/26 and 380/11 nm filters (Semrock, Rochester, NY, USA) in an Oligochrome excitation system (FEI), and fluorescent images were captured using 510/84-nm emission filter (Semrock) with an ORCA-03G digital CCD camera (Hamamatsu, Herrsching am Ammersee, Germany). The 340/380 ratio was used as an index of cytosolic Ca$^{2+}$ levels. Cells were perfused with GSK (10 μM) starting at 30 s and till the end of the experiment.

**TIRF microscopy.** TIRF microscopy was performed using an Observer D1 microscope (Zeiss, Jena, Germany) equipped with a Visitron TIRF system (Visitron Systems, Puchheim, Germany). Cells were viewed using a 100× oil immersion Zeiss TIRF objective and YFP were excited using a 488 nm diode laser. Images were captured using a Prime BSI Express sCMOS camera (Teledyne Photometrics, Tucson, USA) at 515 nm. The distribution and rate of clustering of proteins were analysed using ImageJ 1.53c.

### Statistics
Data analysis and graphical display were performed using Clampfit 11 (Axon Instruments) and GraphPad10 (GraphPad Software, Boston, Massachusetts USA). Data are presented as Mean ± SEM. For normal distributed values (confirmed by Shapiro−Wilk test), Student's two-sample $t$ test or paired $t$ test were used to analyse the statistical significance. Equality of variances was tested by Levene's test and ANOVA was performed when appropriate. For non-normally distributed values, Mann−Whitney rank sum test was applied. In general, differences were considered significant at $P < 0.05$.

### Reporting summary
Further information on research design is available in the Nature Portfolio Reporting Summary linked to this article.

## Data availability
The computational and experimental data generated in this study and the Data Source file have been deposited in the Zenodo data repository under the accession code 10912149 and (https://doi.org/10.5281/

zenodo.11259097) respectively. The PDB entry 6CUD for the cryo-EM structure of human TRPC3 in a lipid-occupied, closed state was used as the starting point to generate the molecular dynamics input structures. The re-entrant loop was modelled using the 5ZBG PDB entry [https://doi.org/10.2210/pdb5ZBG/pdb] of human TRPC3 at 4.36Å resolution. All source data are provided as a Source Data file. Source data are provided with this paper.

## Code availability

For the single channel patch clamp analysis we have created a custom made software for SCANA. Detailed description of the software, code, installation and examples were uploaded to GitHub (https://github.com/medunigraz/scana) and to zenodo.org (https://doi.org/10.5281/zenodo.11203604).

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

## Acknowledgements

This work was supported by the Austrian Science Fund (FWF) through grants P33263 and PAT 9543223 to Klaus Groschner and Thomas Stockner, P35291 to Oleksnadra Tiapko, MEFOgraz and BioTechMed to Oleksandra Tiapko. Hazel Erkan-Candag is a member of PhD program (DK) "Metabolic and Cardiovascular Disease" at the Medical University of Graz (W1226 to Klaus Groschner). We are also grateful to the Vienna Scientific Cluster (VSC) for providing the computational resources for our simulations.

## Author contributions

Conceptualisation, O.T., T.S., K.G.; Methodology, O.T., K.G., T.S.; Software, A.C., T.S., M.G.; Validation, O.T., K.G., T.S.; Formal Analysis, A.C., O.T.; Investigation, A.C., O.T., J.S., P.W., H.E-C.; Resources, T.S., K.G., O.T.; Data Curation, A.C., O.T., K.G., T.S.; Writing - Original Draft, A.C., O.T.; Writing - Review & Editing, A.C., O.T., K.G., T.S.; Visualisation, A.C., O.T.; Supervision, T.S., K.G.; Project Administration, T.S.; Funding Acquisition, T.S., K.G., O.T.

## Competing interests

The authors declare no competing interests.
