## [Peer Review File · Nature Communications]

PIP2 modulates TRPC3 activity via TRP helix and S4-S5 linkerREVIEWER COMMENTS

Reviewer #1 (Remarks to the Author):

In this paper by Clarke and colleagues, from Austria in the Tiapko lab in Graz and the Stockner lab in Vienna, a novel PIP2 sensing site of TRPC3 is identified by Molecular Dynamics (MD) simulations and supported by mutagenesis and electrophysiology experiments. This sensing site involves basic residues in the N-terminal pre-S1 region as well as one residue in the S1 transmembrane helix. Potential links of this novel binding site to a re-entrant loop of the TRP helix and the S4-S5 linker that are critical domains implicated in gating make this a compelling discovery that is likely to be operative in other members of the TRPC subfamily. The study is executed quite well and the data presented on the identification of the novel binding site are of high quality. The suggestions made in this review aim to mainly strengthen the second and more challenging part of the study, namely the allosteric coupling of this PIP2 binding site to determinants of channel gating. Focusing the message and presentation of the work to the main finding, namely the identification of the novel PIP2 binding site, and its connection to critical determinants of gating is needed. The authors should be cautious not to imply that they understand how this novel site couples to the channel gates.

Major points

1) The binding pocket for the PIP2 phosphates is formed by the pre-S1 residues R374, K377, and R380 as well as the S1 residue K385. The authors present evidence in Supp. Fig. 6 that PIP2 reduces the number of hydrogen bonds between K385 and two Asp residues in the re-entrant loop of the TRP helix (D698 and D699) (modeling) and that disabling these interactions reduces the activity of TRPC3 (mutagenesis and electrophysiology). This critical prediction of the model would be strengthened experimentally if the authors were to rescue the significant decreases obtained by either the K385 or the D698 and/or D699 charge reversals in double mutants that may re-establish the interaction(s). The concept is similar to the lovely data the authors presented in Fig. 5 with the R572E – E684R double mutant (DM) that rescued the R572 (S4-S5 linker) and E684 (TRP helix) charge reversals.

2) In lines 459-461, the authors state that “Nonetheless, our simulation data detected only minor contact formation between PIP2 and residues in the L1 region (Supp. Fig. 2a), suggesting that the affinity to the L1 would be much lower”. Before this statement, the authors in lines 455-457 concluded “Our results do not preclude that TRPC3 may be dually regulated by PIP2, involving L3 as well as L1 site as primary recognition sites”. I would reverse the order of these two statements ending this discussion with the statement in lines 455-457 and adding to it that future simulations ought to explore whether these two sites are allosterically coupled by testing the affinities of each site in the presence of PIP2 molecules in each of the two sites.

3) It would strengthen the argument that mutagenesis of the residues involved in coordinating PIP2 in the identified novel site decrease affinity for PIP2. The PH domain experiments are a good control but inside-out macropatches and concentration-response experiments of each of the single mutants of the four residues involved would provide strong evidence of their individual contributions to the overall PIP2 affinity.

Minor points

4) Page 4, lines 133-137: “Moreover, PIP2 rapidly binds to TRPC3, and at higher concentrations ...which showed that Chol typically binds to TRPC3 at regions inaccessible to phospholipids²⁰”. Please show in the supplementary figures the modeling data to substantiate the claims that 10% PIP2 displaces PC, PE and PS.

5) Page 5, lines 154-157: “Moreover, in the presence of the re-entrant loop, the four...enhances it”. It would make the argument stronger if the authors provided as a Supp. Fig. comparable results to Fig. 2g-i on maximal occupancy and contact duration of the D698K/D699K DM.

6) Figure 5e and f, I found confusing referring to the Double Mutant (I would indicated as “DM”) as Control. Please replace “Control” with “DM”.

7) Page 4, line 130: change “can” to “could” in “..., during which PIP2 could laterally diffuse...”

8) In Figure 2h, I, it would help the readers if there was a key (e.g., a line schematic) to put the residue numbers (350...700) in the context of the structurally important domains (e.g., pre-S1, S1, S4-S5 linker, pore domain, re-entrant loop, TRP helix, etc.)

9) Page 6, line 208-209, delete “at” and correct typo “farther” rather than “further” in “...bind here, the majority bind at a site close to the re-entrant loop. A smaller portion binds at a second discrete site slightly farther from the re-entrant loop”.

10) Page 7, line 230: correct the typo from “R374A” to “R380A” in “...directly interacted with PIP2: R374A, K377A, R380A, and K385A (Fig. 3)”.

11) Page 8, line 260; Page 9, line 302; Page 10, line 361, treat “data” as a plural word “these data” and accordingly the verb tense in the sentence.

12) What the authors refer to as “beige” in the figures looks “orange” to my eyes.

13) In general, I would be “more” humble in statements that imply that we now understand how this novel site controls gating. We do appreciate that this is likely an important determinant of activity but we still do not understand how. Please tone down the way lines 472-479 are phrased to stress that the results are consistent with the hypothesis that the distant PIP2 site couples to key elements involved in channel gating to control the activity of this channel.

Reviewer #2 (Remarks to the Author):

The study by Clarke et al. integrates in-silico prediction with experimental validation to identify a significant PIP2 binding site at the pre-S1/S1 nexus of the TRPC3 channel, termed the L3 binding site. This research demonstrates that PIP2 positively modulates TRPC3 function through its interaction at the L3 site. While the study suggests the possibility of dual regulation of TRPC3 by PIP2 at both L3 and L1 sites, similar to the dual role of PIP2 in TRPV1, the simulation data indicate a lower affinity for the L1 site. A key insight provided by this study is the elucidation of a step-by-step mechanism showing how PIP2 binding at the L3 site induces conformational changes that affect the gating structures in the pore domain. The research provides strong evidence of a coordinated movement involving the TRP helix and the S4-S5 linker in this process. Additionally, it reveals the critical role of residue K385, not only as a lipid binding site but also in transmitting the PIP2 binding signal to the TRP helix. Single-channel experiments further support these findings, showing a significant reduction in open channel probability under PIP2 depleted conditions and in the presence of a K385A mutation. Molecular simulations are rigorously carried out appropriately analyzed. The authors' conclusions are well supported by the data and the manuscript is overall clear and detailed enough to ensure reproducibility of the results. I recommend publication of the manuscript after a minor revision:

- The description of the data shown in Supplementary Figure 2 is not sufficiently detailed. The results are relevant as they support the notion that the majority of PIP2 molecules bind to L3. However, the clustering approach used to obtain this result is practically not described. A paragraph with a long and detailed description of these results would improve the clarity of the paper.

Reviewer #3 (Remarks to the Author):

As an essential component of biomembrane, PIP2 is critical for regulation of function, expression, and subcellular localization of diverse membrane proteins. Clarke et al. study the structural biological basis of the modulatory action of PIP2 on the transient receptor potential canonical type 3 (TRPC3) channel, which is recognized to play pivotal roles in regulating neuronal excitability in the brain and pathogenesis of cardiovascular disorders. By employing molecular dynamics simulations, site-directed mutagenesis and patch clamp techniques (both single-channel and whole-cell), authors beautifully identified structural components required for the modulatory action of PIP2 on TRPC3 protein. The experiments are well executed and discussion is sound. My comments are as follows.

1) The results firstly reveal the L3 lipid binding site as a predominant interact site of PIP2 in TRPC3. This finding is comparable to the previous identification of the pre-S1 shoulder as the PIP2 interaction site in the closely related homologue TRPC6 through exhaustive screening cytoplasmic positively charged amino acid residue using mutant analysis with whole-cell patch clamp recording and voltage-dependent phosphatase that hydrolyzes PIP2 in situ (Mori, MX et al. Sci. Rep. 12:10766 (2022)). It can be viewed that the previous work nicely provides the present work with sort of a proof of concept for the molecular dynamics approach taken. An achievement that should be truly appreciated is the identification of multistep propagation of PIP2-induced structural modification from L3 to the pore domain via a salt bridge between the TRP helix and S4-S5 linker. This is only achievable by the authors' molecular dynamics approach. In this sense, authors may consider further discussion that attempts a feedback of the knowledge to interpretation of the phenotypes of the TRPC6 mutants reported in the above previous work, which greatly enhances but never devalues the significance of the present work.

2) Given that TRPC3 plays major roles in certain cardiovascular disease, it is better that this is also mentioned in the Abstract.

3) Page 3, lines 83-87: It is curious whether the influence of masking of negative charges of PIP2, which may concentrate cations nearby channels, by the PH domain (of what?) on TRPC3 currents need to be considered.

4) Page 11, lines 393-400. When the single mutations of R572E and E684R, both of which significantly impairs the activity and modulation of TRPC3, are combined as the double mutation R572E/E684R, the mutant showed restored function. This exciting data should be corroborated by structural analysis using molecular dynamics and the obtained data can be presented as a figure in the revised manuscript.

Point-by-point reply:

First of all, we would like to thank the reviewers for reviewing the manuscript and their constructive criticism. We have addressed the points raised by all three reviewers, as follows:

Reviewer #1 (Remarks to the Author):

1) The binding pocket for the PIP₂ phosphates is formed by the pre-S1 residues R374, K377, and R380 as well as the S1 residue K385. The authors present evidence in Supp. Fig. 6 that PIP₂ reduces the number of hydrogen bonds between K385 and two Asp residues in the re-entrant loop of the TRP helix (D698 and D699) (modeling) and that disabling these interactions reduces the activity of TRPC3 (mutagenesis and electrophysiology). This critical prediction of the model would be strengthened experimentally if the authors were to rescue the significant decreases obtained by either the K385 or the D698 and/or D699 charge reversals in double mutants that may reestablish the interaction(s). The concept is similar to the lovely data the authors presented in Fig. 5 with the R572E – E684R double mutant (DM) that rescued the R572 (S4-S5 linker) and E684 (TRP helix) charge reversals.

Response: The electrophysiology data shows that mutations of negatively charged residues on the TRP helix (D698K/D699K) and D698I/D699K) as well as positively charged residue K701, have no effect on channel activity (Supp. Figure 9). As a result, we concluded that the side chains of D698, D699 and K701 are not involved in signal propagation. To confirm this, we created a coarse-grained double mutant (D698K/D699K) and ran 5 repeats of a 20 μs simulation. The subsequent analysis is included in Supp. Figure 9d-f. It confirms that PIP₂ still accumulates at the L3 at levels comparable to that of the WT channel. This is consistent with the experimental data showing no functional effects. We propose that K385 interacts with the negative helical dipole of the TRP helix through the carbonyl backbone atoms of D698 and D699, which is not affected by charge modifying mutations. Thus, the charge reversal mutations approach does not provide useful information.

2) In lines 459-461, the authors state that “Nonetheless, our simulation data detected only minor contact formation between PIP₂ and residues in the L1 region (Supp. Fig. 3a), suggesting that the affinity to the L1 would be much lower”. Before this statement, the authors in lines 455-457 concluded “Our results do not preclude that TRPC3 may be dually regulated by PIP₂, involving L3 as well as L1 site as primary recognition sites”. I would reverse the order of these two statements ending this discussion with the statement in lines 455-457 and adding to it that future simulations ought to explore whether these two sites are allosterically coupled by testing the affinities of each site in the presence of PIP₂ molecules in each of the two sites.

Response: We reordered the sentences according to the suggestion, which makes the statement much clearer. We have also added an additional sentence to discuss the possible future direction of the research.

3) It would strengthen the argument that mutagenesis of the residues involved in coordinating PIP₂ in the identified novel site decrease affinity for PIP₂. The PH domain experiments are a good control but inside-out macropatches and concentration response experiments of each of the single mutants of the four residues involved would provide strong evidence of their individual contributions to the overall PIP₂ affinity.

Response: We thank the reviewer for this valuable suggestion. Accordingly, we conducted inside-out patch clamp recordings from TRPC3, K377A, R380A, and K385A to compare the behaviour of TRPC3 WT with that of mutant channels during exposure to the PIP₂ scavenger PalPIP2 (results are summarised in **Suppl. Figures 7 and 8**). TRPC3 WT, as well as K377A and R380A, exhibited a more pronounced decline (run-down) in channel activity (N^*Po) in the presence of PalPIP2 (10 μ M; **Suppl. Figure 8a**) as compared to controls (DMSO; **Suppl. Figure 8b**). Importantly, we observed stable activity (no significant decline in N^*Po) for the K385A mutant in the same setting (**Suppl. Figure 8a-b**). Our whole-cell recordings from TRPC3-overexpressing HEK293 cells revealed that PalPIP2 concentrations of >10 μ M are required to affect the activity of TRPC3 WT channels (**Suppl. Figure 7a**). Using this PalPIP2 concentration the inside-out recording configuration, we observed a slightly but insignificantly faster rundown in K377A or R380A mutants as compared to WT (**Figure below**) This demonstrates the minor if any impact of these individual charges for the affinity of L3 to bind PIP₂. The L3 site is a highly positively charged pocket that includes several positively charged residues. Consequently, we assume that the overall positive charge density in L3 remains sufficiently high for PIP₂ binding. Specifically, residues adjacent to K377 and R380 can compensate. Similar redundancy has been observed in other membrane proteins regulated by PIP₂, such as the serotonin transporter SERT¹.

Most importantly, our results obtained from the inside-out patch clamp approach clearly confirm that K385A is essential for productive sensing of PIP₂ binding to L3 by the channel (**Figure 4 and Suppl. Figure 8**).

We excluded the R374A residue from the experiments due to its lowest binding probability for PIP₂ according to the results from our MD simulations (**Figure 2f**).

1

¹ 1. Buchmayer, F. et al. Amphetamine actions at the serotonin transporter rely on the availability of phosphatidylinositol-4,5-bisphosphate. *Proc.*

PaPIP2

*In this figure we show a summary of the linear regression plotting of the N^*Po rundown from inside-out patches in presence of PaPIP2 (combined data from Suppl. Figure 7c). Data is presented as Mean \pm SEM. Statistical significance was assessed using ANOVA followed by a two-tailed multiple t test with Bonferroni correction. ** $P < 0.01$, ns = not significant*

4) Page 4, lines 133-137: “Moreover, PIP₂ rapidly binds to TRPC3, and at higher concentrations ...which showed that Chol typically binds to TRPC3 at regions inaccessible to phospholipids²⁰”. Please show in the supplementary figures the modeling data to substantiate the claims that 10% PIP₂ displaces PC, PE and PS.

Response: *We have now included new modelling data as **Suppl. Figure 2** to confirm this claim. Critically, this data shows that only PS is significantly reduced in the presence of 10% PIP₂. As a result, we have now amended our sentence in the main text of the manuscript to read: “Moreover, PIP₂ rapidly binds to TRPC3, and at higher concentrations (10%, Fig. 1e) it displaces the negatively charged lipid phosphatidylserine (PS) (**Suppl. Figure 2**). PIP₂ does not appear to displace phosphatidylcholine (PC) or phosphatidylethanolamine (PE) (**Suppl. Figure 2**). It also does not appear to compete with cholesterol (Chol) for binding to TRPC3”.*

5) Page 5, lines 154-157: “Moreover, in the presence of the re-entrant loop, the four...enhances it”. It would make the argument stronger if the authors provided as a Suppl. Fig. comparable results to Fig. 2g-i on maximal occupancy and contact duration of the D698K/D699K DM.

Response: *We have now created a coarse-grained double mutant (D698K/D699K) and ran 5 independent simulations, each 20 μ s long simulation to explore the effect of the double*

mutant on PIP₂ recruitment to the L3 site. This data is now included in **Suppl. Figure 9**. Maximum occupancy and contact duration analysis show that the L3 site (composed of the pre-S1, S1, and re-entrant loop) is still associated with the highest maximum occupancy and contact duration values. Density map analysis confirms that PIP₂ accumulates at the L3. There appear to be minor changes associated with the mutation, such as a slight increase in PIP₂ binding to the S4-S5 linker; however, given that the majority of PIP₂ binds to the L3 as in WT, we believe our data is consistent with the experimental data which shows no change in net current density between WT and mutant.

The new data is included in **Suppl. Figure 9** and we have also included the sentence “Moreover, a coarse-grained computational mutant D698K/D699K accumulated PIP₂ at the L3 site at levels comparable to the WT protein (**Suppl. Fig. 9d-f**).” in our results section.

6) Figure 5e and f, I found confusing referring to the Double Mutant (I would indicated as “DM”) as Control. Please replace “Control” with “DM”.

Response: We have now amended the figure.

7) Page 4, line 130: change “can” to “could” in “..., during which PIP₂ could laterally diffuse...”.

Response: We have made this change.

8) In Figure 2h, I, it would help the readers if there was a key (e.g., a line schematic) to put the residue numbers (350...700) in the context of the structurally important domains (e.g., pre-S1, S1, S4-S5 linker, pore domain, re-entrant loop, TRP helix, etc.).

*Response: In this figure, the vertical blocks indicate the transmembrane helices of TRPC3, and they are coloured according to the structurally important domains, as in **Figure 2e**. However, for additional clarity, we have now also included the helix naming above the diagram.*

9) Page 6, line 208-209, delete “at” and correct typo “farther” rather than “further” in “...bind here, the majority bind at a site close to the re-entrant loop. A smaller portion binds at a second discrete site slightly farther from the re-entrant loop”.

Response: We thank the reviewer for noticing these mistakes.

10) Page 7, line 230: correct the typo from “R374A” to “R380A” in “...directly interacted with PIP₂: R374A, K377A, R380A, and K385A (Fig. 3)”.

Response: Please accept our apologies for this error, which was corrected. We thank the reviewer for noticing this mistake.

11) Page 8, line 260; Page 9, line 302; Page 10, line 361, treat “data” as a plural word “these

data” and accordingly the verb tense in the sentence.

Response: Please accept our apologies for this error, which was corrected. We thank the reviewer for noticing this mistake.

12) What the authors refer to as “beige” in the figures looks “orange” to my eyes.

Response: We agree and have changed the description accordingly.

13) In general, I would be “more” humble in statements that imply that we now understand how this novel site controls gating. We do appreciate that this is likely an important determinant of activity but we still do not understand how. Please tone down the way lines 472-479 are phrased to stress that the results are consistent with the hypothesis that the distant PIP₂ site couples to key elements involved in channel gating to control the activity of this channel.

Response: We agree with the reviewer’s notion and tuned down our conclusion as follows: “Our study unveiled several critical steps of the mechanism by which PIP₂ binding to the L3 site is translated into conformational changes of gating elements in the pore domain to govern channel activity. We present evidence supporting a multistep regulatory process involving coordinated movement of the TRP helix and the S4-S5 linker. Additionally, our findings identify K385 as a part of a lipid binding site and as a pivotal element transmitting the PIP₂ binding signal to the TRP helix within the signal transduction pathway.”.

Reviewer #2 (Remarks to the Author):

The study by Clarke et al. integrates in-silico prediction with experimental validation to identify a significant PIP₂ binding site at the pre-S1/S1 nexus of the TRPC3 channel, termed the L3 binding site. This research demonstrates that PIP₂ positively modulates TRPC3 function through its interaction at the L3 site. While the study suggests the possibility of dual regulation of TRPC3 by PIP₂ at both L3 and L1 sites, similar to the dual role of PIP₂ in TRPV1, the simulation data indicate a lower affinity for the L1 site. A key insight provided by this study is the elucidation of a step-by-step mechanism showing how PIP₂ binding at the L3 site induces conformational changes that affect the gating structures in the pore domain. The research provides strong evidence of a coordinated movement involving the TRP helix and the S4-S5 linker in this process. Additionally, it reveals the critical role of residue K385, not only as a lipid binding site but also in transmitting the PIP₂ binding signal to the TRP helix. Single-channel experiments further support these findings, showing a significant reduction in open channel probability under PIP₂ depleted conditions and in the presence of a K385A mutation. Molecular simulations are rigorously carried out appropriately analyzed. The authors' conclusions are well supported by the data and the manuscript is overall clear and detailed enough to ensure reproducibility of the results. I recommend publication of the manuscript after a minor revision:

- The description of the data shown in Supplementary Figure 2 is not sufficiently detailed. The results are relevant as they support the notion that the majority of PIP₂ molecules bind to L3. However, the clustering approach used to obtain this result is practically not described. A paragraph with a long and detailed description of these results would improve the clarity of the paper.

*Response: We agree that **Suppl. Figure 2** (in the revised version it is **Suppl. Figure 3**) was poorly explained in the original manuscript. We have now included an expanded explanation in the Figure legend and also included an additional section in the Methods section. We have also amended the figure slightly to improve the clarity. We thank the reviewer for the comments.*

Reviewer #3 (Remarks to the Author):

As an essential component of biomembrane, PIP₂ is critical for regulation of function, expression, and subcellular localization of diverse membrane proteins. Clarke et al. study the structural biological basis of the modulatory action of PIP₂ on the transient receptor potential canonical type 3 (TRPC3) channel, which is recognized to play pivotal roles in regulating neuronal excitability in the brain and pathogenesis of cardiovascular disorders. By employing molecular dynamics simulations, site-directed mutagenesis and patch clamp techniques (both single-channel and whole-cell), authors beautifully identified structural components required for the modulatory action of PIP₂ on TRPC3 protein. The experiments are well executed and discussion is sound. My comments are as follows.

1) The results firstly reveal the L3 lipid binding site as a predominant interact site of PIP₂ in TRPC3. This finding is comparable to the previous identification of the pre-S1 shoulder as the PIP₂ interaction site in the closely related homologue TRPC6 through exhaustive screening cytoplasmic positively charged amino acid residue using mutant analysis with whole-cell patch clamp recording and voltage-dependent phosphatase that hydrolyzes PIP₂ in situ (Mori, MX et al. Sci. Rep. 12:10766 (2022)). It can be viewed that the previous work nicely provides the present work with sort of a proof of concept for the molecular dynamics approach taken. An achievement that should be truly appreciated is the identification of multistep propagation of PIP₂-induced structural modification from L3 to the pore domain via a salt bridge between the TRP helix and S4-S5 linker. This is only achievable by the authors' molecular dynamics approach. In this sense, authors may consider further discussion that attempts a feedback of the knowledge to interpretation of the phenotypes of the TRPC6 mutants reported in the above previous work, which greatly enhances but never devalues the significance of the present work.

Response: We thank the reviewer for the suggestion. The work from Mori et al. nicely complements our present study and we are happy to include a more thorough discussion of this issue: "This localisation of a regulatory PIP₂ binding is consistent with a site recently proposed for the highly homologous TRPC6 channel (Mori et al, 2022). Significantly, in this study, a K442Q mutation was found to result in a decrease of PIP₂ binding affinity about 5- to 8-fold as compared to WT TRPC6 (Mori et al, 2022). K442 is identical to the K385 residue found in TRPC3, which we here propose to function both as a PIP₂ binding residue and a transducer of the PIP₂ binding signal. Interestingly, the Mori et al study (2022), also proposes a role for residues of the distal TRP box, such as K781 and K782, in PIP₂ sensing. Due to lack of structural information of this region in the cryo-EM structures, we have been unable to explore whether this region plays a role in PIP₂ sensing by TRPC3. However, with the advent of de novo structure prediction methods such as AlphaFold, this may successfully be addressed by future research."

2) Given that TRPC3 plays major roles in certain cardiovascular disease, it is better that this is also mentioned in the Abstract.

Response: As TRPC3 channels are considered promising targets in the treatment of cardiovascular diseases, we agree that this important point should be mentioned in the abstract. We have now included the sentence: “These structural insights into the function of TRPC3 are invaluable for understanding the role of the TRPC subfamily in health and disease in native tissue, in particular for cardiovascular diseases, in which TRPC3 channels play a major role.”

3) Page 3, lines 83-87: It is curious whether the influence of masking of negative charges of PIP₂, which may concentrate cations nearby channels, by the PH domain (of what?) on TRPC3 currents need to be considered.

Response: We thank the reviewer for the interesting question. PIP₂ is indeed known to strongly interact with Ca²⁺ and, therefore masking PIP₂ may in principle impact on local free Ca²⁺. In this respect it is important to consider that the PIP₂ concentration in the inner leaflet of cells amounts to about 2% and other negatively charged lipids do significantly contribute to overall binding of Ca²⁺ to the membrane. Nonetheless, PIP₂ scavenging may be considered to exert some effect on local divalent availability, even though PIP₂ is not the sole negatively charged lipid in the inner membrane leaflet. Moreover, Guo et al. (2022) provided evidence that regulatory coordination of Ca²⁺ ions within the TRPC3 complex involves multiple sites, which are primarily reached by Ca²⁺ originating from the extracellular space, passing through the pore of the channel. Therefore, masking the negative charges of inner leaflet PIP₂ is unlikely to exert a marked impact on Ca²⁺-dependent regulation. Our experimental conditions, using 2 mM extracellular Ca²⁺ combined with internal divalent buffers (EGTA in whole cell and inside-out patch clamp; compositions are described in the Methods section) are expected to provide rather stable Ca²⁺ concentrations at the cytoplasmic face of the channel. Moreover, we found that the PIP₂-insensitive mutant K385A displayed unchanged Ca²⁺ permeability (**Suppl. Figure 6b**). We assume that Ca²⁺ flux through the channel is the primary determinant of Ca²⁺-dependent regulation of the TRPC3 channel overriding possible minor effects that may arise from altered divalent availability at the membrane surrounding the channel due to PIP₂ scavenging.

4) Page 11, lines 393-400. When the single mutations of R572E and E684R, both of which significantly impairs the activity and modulation of TRPC3, are combined as the double mutation R572E/E684R, the mutant showed restored function. This exciting data should be corroborated by structural analysis using molecular dynamics and the obtained data can be presented as a figure in the revised manuscript.

Response: We thank the reviewer for this suggestion. To corroborate the experimental data, we created a R572E/E684R double mutant and conducted 150 ns long all atom simulation. We then calculated the distances between residues R572E/E684R as in **Figure 5b**. The data is presented in **Supplementary Figure 10d**. It shows that the mutant channel retains a modest fraction of residues forming a salt bridge. We believe this is consistent with the experimental charge-swapped mutant, which showed a modest regain in function. We have now included this data in **Suppl. Figure 10d** and added the sentence to the manuscript “A computational all-atom simulation of the mutant R572E/E684R also retained a modest

fraction where these two residues formed a salt bridge (Supplementary Fig. 10d) during the course of a 150 ns simulation.” in the Results section of the manuscript.

REVIEWERS' COMMENTS

Reviewer #1 (Remarks to the Author):

I do feel that the authors were very responsive to my comments and I feel satisfied and comfortable to endorse the paper for publication to NCOMMS. I find the revised manuscript improved and the study worthy of the precious space of NCOMMS.

Diomedes Logothetis

Reviewer #2 (Remarks to the Author):

The authors have properly addressed my concerns, I recommend publication of the manuscript in the present form.

Reviewer #3 (Remarks to the Author):

This interesting paper has been revised appropriately in response to the comments by the reviewers.